# Learning to Generate Stylized Handwritten Text via a Unified Representation of Style, Content, and Noise

**Honglie Wang**[1,2]**, Yan-Ming Zhang**[1,2]**, Wangzi Yao**[1,2]**, Fei Yin**[1,2]**, Cheng-Lin Liu**[1,2]

[1]School of Artificial Intelligence, University of Chinese Academy of Sciences,
  Beijing 100049, China
[2]State Key Laboratory of Multimodal Artificial Intelligence Systems (MAIS),
  Institution of Automation Chinese Academy of Sciences,
  Beijing 100190, China
{wanghonglie2023, yaowangzi2023}@ia.ac.cn
{ymzhang, fyin, liucl}@nlpr.ia.ac.cn

## Abstract

Handwritten Text Generation (HTG) seeks to synthesize realistic and personalized handwriting by modeling stylistic and structural traits. While recent diffusion-based approaches have advanced generation fidelity, they typically rely on auxiliary style or content encoders with handcrafted objectives, leading to complex training pipelines and limited interaction across factors. In this work, we present InkSpire, a diffusion transformer based model that unifies style, content, and noise within a shared latent space. By eliminating explicit encoders, InkSpire streamlines optimization while enabling richer feature interaction and stronger in-context generation. To further enhance flexibility, we introduce a multi-line masked infilling strategy that allows training directly on raw text-line images, together with a revised positional encoding that supports arbitrary-length multi-line synthesis and fine-grained character editing. Moreover, InkSpire is trained on a bilingual Chinese–English corpus, enabling a single model to handle both Chinese and English handwriting generation with high fidelity and stylistic diversity, thereby overcoming the need for language-specific systems. Extensive experiments on IAM and ICDAR2013 demonstrate that InkSpire achieves superior structural accuracy and stylistic diversity compared to prior state-of-the-art methods.

## 1 Introduction

Handwritten Text Generation (HTG) aims to synthesize realistic and personalized handwriting from arbitrary digital input by modeling traits such as slant, cursiveness, and stroke dynamics. Leveraging the scalability of modern models, HTG enables human-like handwriting generation with broad applications in assistive technology, personalized rendering, font design, historical manuscript restoration, and writer identification.

Diffusion models have recently emerged as the dominant paradigm for offline handwritten text synthesis, surpassing GAN-based approaches in generation quality. Early attempts at style conditioning (e.g., WordStylist (Nikolaidou et al., 2023), GC-DDPM (Ding et al., 2023), CTIG-DM (Zhu et al., 2023)) relied on fixed writer IDs, which constrained stylistic diversity. Subsequent methods introduced dedicated style encoders trained with tailored objectives to capture broader stylistic variations. For instance, One-DM (Dai et al., 2024) employs a Laplacian contrastive loss to emphasize fine-grained features, while DiffusionPen (Nikolaidou et al., 2024) integrates triplet and classification losses to enhance style discrimination. With respect to content fidelity, TGC-Diff (Wang et al., 2025) proposes a high-frequency mask loss to preserve structural details. While these approaches improve model performance, they still handle style, content, and noise as separate factors, each constrained by manually crafted losses, which increases the difficulty of optimization.

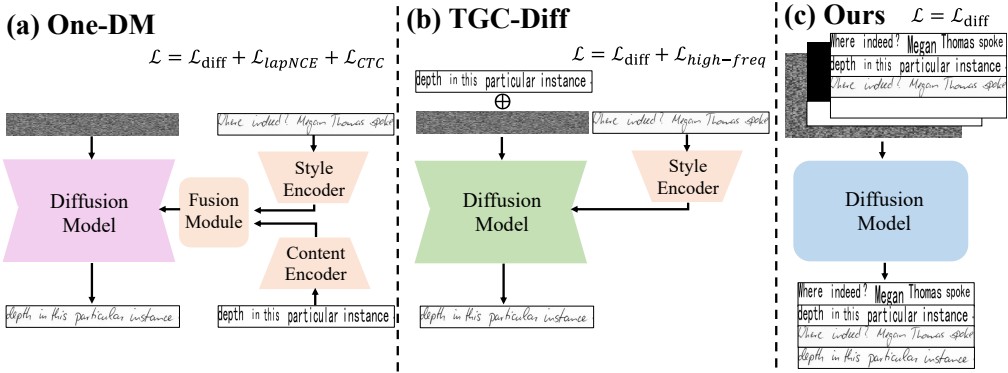

Figure 1: Structural comparison of handwritten text-line generation methods. Unlike previous approaches, our model integrates style and content modeling without additional encoders, yielding a streamlined architecture.

To efficiently integrate diffusion models into HTG systems, it is crucial to examine the role of auxiliary style and content encoders. As illustrated in Figure 1(a), One-DM fuses style and content features to provide useful guidance for style learning but fails to capture fine-grained structures and spatial information. In contrast, TGC-Diff in Figure 1(b) constructs content features within the same latent space as noise, facilitating a seamless connection between content and noise representations and enhancing structural fidelity and spatial consistency. If content and noise can be jointly represented in a shared latent space, is it possible to design a single unified diffusion model that simultaneously processes style, content, and noise? Such a framework would not only obviate the need for redundant encoders with complex handcrafted losses, but also improve performance by enabling efficient interaction within a common latent representation.

Building on this insight, we introduce InkSpire, a diffusion transformer model for stylized handwriting generation, which is "inspired" by in-context latent "ink" tokens. As illustrated in Figure 1(c), InkSpire leverages a shared latent space for style, content, and noise, replacing separate encoders with a streamlined architecture that facilitates effective feature interaction. Moreover, recent advances in large text-to-image diffusion models have demonstrated strong in-context generation capabilities. To apply this unified modeling ability to our HTG task, we design a multi-line masked infilling strategy for training and remove the text encoder to enable purely visual conditioning. In addition, we revise the positional encoding mechanism to support multi-line generation of arbitrary length and train the model on a mixed Chinese–English dataset, thereby enabling bilingual stylized handwriting synthesis. By integrating these innovations, InkSpire achieves high-fidelity and stylistically diverse handwriting generation and editing, while streamlining the overall training process.

In summary, the main contributions of this paper are as follows:

- We propose InkSpire, a novel handwriting generation framework that unifies the modeling of style, content, and noise without relying on explicit style or content encoders. By leveraging in-context generation with diffusion transformer models, InkSpire simplifies the training pipeline while preserving high fidelity and stylistic diversity.

- We introduce a multi-line masked infilling strategy that enables the model to be trained directly on raw multi-line text images, without requiring complex data preprocessing. Together with a revised positional encoding scheme, InkSpire supports the generation of multi-line handwriting of arbitrary length as well as fine-grained, character-level editing.

- We enable bilingual handwriting generation within a single model by training InkSpire on a mixed Chinese–English dataset. This design allows the synthesis of high-quality handwritten text in both English and Chinese scripts, thereby overcoming the constraints of language-specific handwriting systems.

- We conduct extensive experiments on the ICDAR2013 and IAM datasets, demonstrating that InkSpire produces handwriting with superior structural accuracy and stylistic consistency, outperforming other methods in both qualitative and quantitative evaluations.

## 2 RELATED WORK

Handwritten data is commonly categorized into two modalities: online trajectory sequences and offline static images. Online handwriting, which captures the dynamic pen trajectory during the writing process, has been widely studied with various generative models, including RNN-based approaches (Kotani et al., 2020; Zhao et al., 2020; Zhang et al., 2017), Transformer-based architectures (Dai et al., 2023), and diffusion-based methods (Luhman & Luhman, 2020; Ren et al., 2023). In contrast, offline handwriting represents static visual appearances, conveying natural characteristics such as stroke thickness, curvature, and ink density.

### 2.1 OFFLINE HANDWRITING GENERATION

Early approaches to offline handwriting synthesis predominantly relied on Generative Adversarial Networks (Alonso et al., 2019; Xie et al., 2021; Gan & Wang, 2021; Kong et al., 2022; Liu et al., 2022), where adversarial training was used to generate visually plausible text. Later, transformer-based models such as HWT (Bhunia et al., 2021) and VATr (Pippi et al., 2023a) introduced hybrid CNN–transformer designs that enhanced style representation learning and improved generalization.

More recently, diffusion models have emerged as the dominant paradigm, synthesizing handwriting through iterative denoising with fine-grained control over style and content. State-of-the-art methods (Wang et al., 2025; Pippi et al., 2025; Dai et al., 2024; Nikolaidou et al., 2023; Ding et al., 2023; Zhu et al., 2023) demonstrate strong style adaptation from only a few references, enabling diverse and personalized handwriting generation with minimal supervision.

### 2.2 IN-CONTEXT GENERATION

Recent advances in diffusion-based generative modeling increasingly adopt in-context generation for controllable and personalized image synthesis. Early works such as InstructPix2Pix (Brooks et al., 2023) and its successors (Boesel & Rombach, 2024) fine-tuned diffusion models with synthetic instruction–response pairs for diverse editing tasks, marking initial attempts to align generation with user intent. Subsequently, instruction-driven editors including Emu Edit (Sheynin et al., 2024), OmniGen (Xiao et al., 2025), HiDream-I1 (Cai et al., 2025), ICEdit (Zhang et al., 2025), JRM (Yang et al., 2026) and SpatialReward (Long et al., 2026) leveraged refined datasets and task-specific architectural enhancements to improve alignment and fidelity. Huang et al. (2024) extend this paradigm by introducing task-specific LoRA branches within diffusion transformers. Although unified models for both editing and generation have been explored in printed document understanding (Tang et al., 2023; Chen et al., 2023), our approach is the first to bring in-context modeling strength of unified editing-and-generation framework into the handwriting domain.

## 3 METHODS

### 3.1 PRELIMINARY

#### 3.1.1 DATA NOTION

The multi-line handwritten dataset is represented as a collection of transcribed image-text pairs $\{(\mathbf{X}, \mathbf{C})\}$, where $\mathbf{X}$ denotes a stylized text-line image and $\mathbf{C}$ denotes the corresponding textual content. Let $\mathbf{C} = (c^1, c^2, \cdots, c^n)$ denote a text sequence of length $n$, where each element corresponds to either a character in logographic writing systems (e.g., Chinese) or a word in phonemic writing systems (e.g., English).

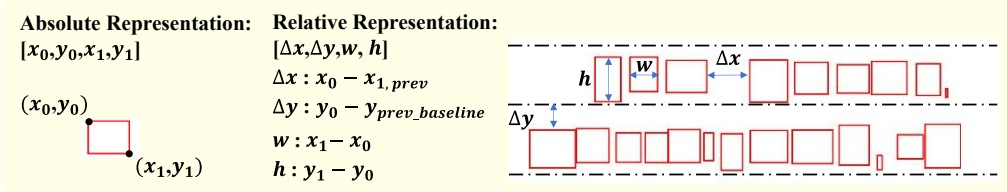

Figure 2: This figure illustrates the transformation from absolute to relative positional representations in layout information. All positions are normalized by the original paragraph image width.

### 3.1.2 TASK ANALYSIS

The objective of handwritten text-line generation is to synthesize text-line images that accurately convey the given textual content $\mathbf{C}$ while faithfully imitating the writing style of a target author, as specified by style reference samples $\mathbf{X}_s$. Formally, the task can be expressed as modeling the conditional distribution: $p(\mathbf{X} \mid \mathbf{C}, \mathbf{X}_s)$.

However, directly modeling this distribution is challenging due to the lack of explicit character positions. Following prior work Wang et al. (2025); Yao et al. (2025), we introduce a content image $\mathbf{X}_c$, where each character in the text-line image $\mathbf{X}$ is replaced with a glyph rendered in a standard font according to the stylized layout. This allows us to model the joint distribution of the text-line image and its content image: $p(\mathbf{X}, \mathbf{X}_c \mid \mathbf{C}, \mathbf{X}_s)$. In line with Ren et al. (2023), the joint distribution can be factorized as:

$$p(\mathbf{X}, \mathbf{X}_c \mid \mathbf{C}, \mathbf{X}_s) = p(\mathbf{X}_c \mid \mathbf{C}, \mathbf{X}_s) \, p(\mathbf{X} \mid \mathbf{C}, \mathbf{X}_s, \mathbf{X}_c) \\ = p(\mathbf{X}_c \mid \mathbf{C}, \mathbf{X}_s) \, p(\mathbf{X} \mid \mathbf{X}_s, \mathbf{X}_c). \tag{1}$$

Accordingly, the overall generation process decomposes into two components: (i) layout generation model $p(\mathbf{X}_c \mid \mathbf{C}, \mathbf{X}_s)$, and (ii) image generation model $p(\mathbf{X} \mid \mathbf{X}_s, \mathbf{X}_c)$.

### 3.2 LAYOUT GENERATION

Given an input character sequence $\mathbf{C} = (c^1, c^2, \cdots, c^n)$, we define a corresponding layout sequence $\mathbf{B} = (b^1, b^2, \cdots, b^n)$, where each $b^i$ denotes the bounding box of character $c^i$. Specifically, $b^i$ is parameterized by four normalized values that capture the character's [width $w$, height $h$, horizontal offset from the previous character $\Delta x$, vertical offset from the baseline of the preceding text-line $\Delta y$], as illustrated in Figure 2. We explore three strategies for modeling layout representations:

- Autoregressive Modeling

$$p(b^i \mid [(b^1, c^1), (b^2, c^2), ..., (b^{i-1}, c^{i-1})], c^i, \mathbf{X}_s). \tag{2}$$

- Masked Layout Modeling

$$p(b^i, i \in \mathcal{M} \mid [(b^j, c^j), j \notin \mathcal{M}], [c^k, k \in \mathcal{M}], \mathbf{X}_s). \tag{3}$$

- Masked Modeling with Conditional Flow Matching

$$p(b^i_{t-1}, i \in \mathcal{M} \mid [(b^j, c^j), j \notin \mathcal{M}], [(b^k_t, c^k), k \in \mathcal{M}], \mathbf{X}_s), \tag{4}$$

where $\mathcal{M}$ denotes the index set of masked positions and $t$ denotes the timestep.

Specifically, the autoregressive model employs several transformer decoder layers, while both the masked layout model and its CFM variant utilize several transformer encoder layers. During training, the autoregressive and masked layout models are optimized by minimizing the average L1 distance between the predicted and ground-truth bounding box parameters (width, height, horizontal offset, and vertical offset) for the relevant characters. For the CFM model, the training loss

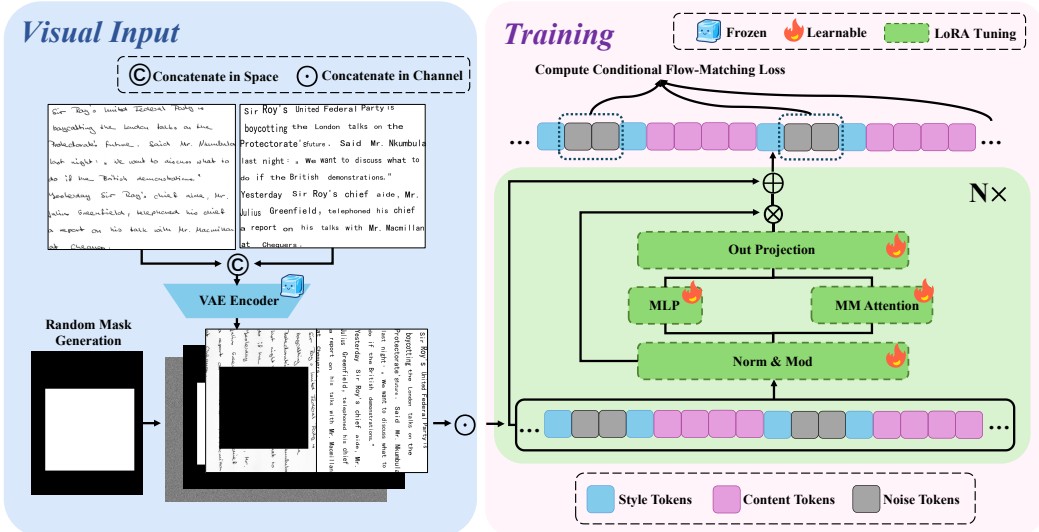

Figure 3: Overview of InkSpire. Our model achieves unified modeling by simply concatenating $\mathbf{X}$ with $\mathbf{X}_c$ and training directly on multi-line images with mask noise prediction. No additional encoders or extra designed loss are needed.

is defined as the average L1 distance between the predicted and ground-truth velocity of these four layout parameters at each timestep. After training, we render the content image $\mathbf{X}_c$ based on the predicted layout sequence $\mathbf{B}$ and a chosen standard font, thereby successfully modeling the conditional distribution $p(\mathbf{X}_c \mid \mathbf{C}, \mathbf{X}_s)$.

## 3.3 IMAGE GENERATION

### 3.3.1 MULTI-LINE MASKED INFILLING STRATEGY

Leveraging the layout generation model, the original image-text pairs $\{(\mathbf{X}, \mathbf{C})\}$ are transformed into image-content pairs $\{(\mathbf{X}, \mathbf{X}_c)\}$. Previous approaches typically construct paired target and style images $\{(\mathbf{X}_{tar}, \mathbf{X}_s)\}$, by cropping two distinct text-lines from a single author's multi-line image $\mathbf{X}$. Images are then resized to a fixed height for training convenience. Under this setup, diffusion models are trained to learn $p(\mathbf{X}_{tar,t-1} \mid \mathbf{X}_{tar,t}, \mathbf{X}_s, \mathbf{X}_c)$, where $\mathbf{X}_c$ or $\mathbf{X}_s$ are typically processed by separate encoders, and thus do not share the same feature space as $\mathbf{X}_{tar,t}$. Such preprocessing is suboptimal as it (i) overly shrinks characters in highly slanted lines, (ii) introduces inconsistent distortions across lines with different slants, and (iii) discards inter-line style cues, thereby limiting generative capability and hindering resolution generalization.

To enable direct training on the original image pairs $\{(\mathbf{X}, \mathbf{X}_c)\}$, we first randomly crop fixed-size patches of size $P \times P$ from the original handwritten page images, where $P$ is a hyperparameter controlling the patch size. On these patches, a random binary mask image $\mathbf{M}$ of the same size is applied. This mask partitions the image into two complementary components: the masked region $\mathbf{X}_{mis} = \mathbf{M} \otimes \mathbf{X}$ and the observed context $\mathbf{X}_{ctx} = (\mathbf{1} - \mathbf{M}) \otimes \mathbf{X}$. With this construction, there is no longer a need to explicitly crop paired samples $\{(\mathbf{X}_{tar}, \mathbf{X}_s)\}$ from $\mathbf{X}$. In fact, $\mathbf{X}_{mis}$ implicitly corresponds to $\mathbf{X}_{tar}$, while $\mathbf{X}_{ctx}$ serves the role of $\mathbf{X}_s$. Consequently, the training objective is reformulated to $p(\mathbf{X}_{mis,t-1} \mid \mathbf{X}_{mis,t}, \mathbf{X}_{ctx}, \mathbf{X}_c)$, thereby eliminating the need for additional preprocessing.

Under this probabilistic modeling framework, style, content, and noise can be jointly represented within a unified latent space. As illustrated in Figure 3, we begin by constructing the image $\mathbf{I}$ through the spatial concatenation of $\mathbf{X}$ and $\mathbf{X}_c$. A random binary mask image $\mathbf{I}_m$ is then applied to generate the masked input $\mathbf{I}_i$. After encoding with the VAE encoder and applying patchification, we obtain the masked image tokens $\mathbf{F}_i$ together with the mask tokens $\mathbf{F}_m$. Finally, the noisy image tokens $\mathbf{F}_n$ are concatenated with $\mathbf{F}_i$ and $\mathbf{F}_m$ along the channel dimension. The overall procedure

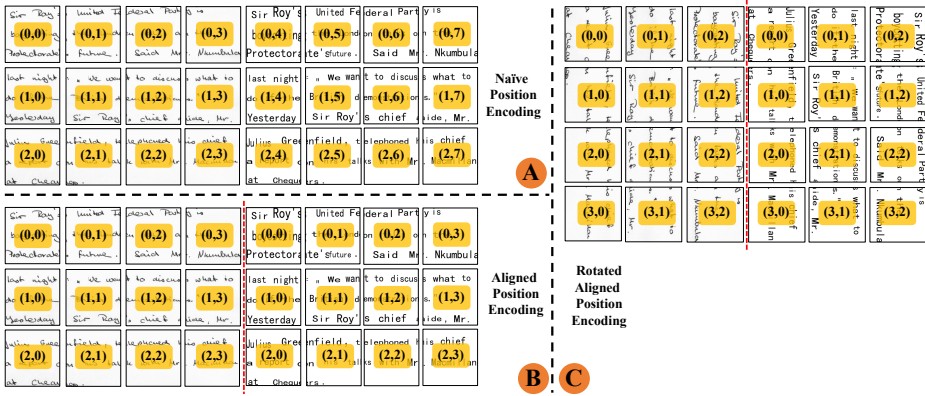

Figure 4: Comparison of different positional encoding strategies. We introduce Aligned Position Encoding (APE) to better guide the spatial layout of generated handwritten text, and further propose a variant, R-APE, tailored for long text lines.

can be formally expressed as follows:

$$\mathbf{I} = \mathbf{X} \copyright \mathbf{X}_c, \tag{5}$$

$$\mathbf{I}_i = \mathbf{I} \otimes (\mathbf{1} - \mathbf{I}_m), \tag{6}$$

$$\mathbf{F}_i = \text{Patchify}(\text{VAE}(\mathbf{I}_i)), \tag{7}$$

$$\mathbf{F}_m = \text{Patchify}(\mathbf{I}_m), \tag{8}$$

$$\mathbf{F}_{input} = \mathbf{F}_n \odot \mathbf{F}_i \odot \mathbf{F}_m. \tag{9}$$

Here, $\copyright$ denotes the concatenation operation along the spatial dimension, $\otimes$ represents the element-wise (Hadamard) product, $\odot$ denotes the concatenation operation along the channel dimension, $\mathbf{F}_i$ encapsulates the information from both $\mathbf{X}_{ctx}$ and $\mathbf{X}_c$. Since the same VAE encoder is utilized, style, content, and noise are jointly represented within a unified feature space.

### 3.3.2 MASKED CONDITIONAL FLOW-MATCHING OBJECTIVE

We employ a flow-matching training objective to optimize the model. Specifically, given a clean latent code $\mathbf{x}_0$, a Gaussian noise sample $\mathbf{z}_1 \sim \mathcal{N}(0, I)$, and a time-dependent noise scale $\sigma_t$, we generate the noisy latent input $\mathbf{x}_0$ via a convex combination:

$$\mathbf{x}_t = (1 - \sigma_t)\mathbf{x}_0 + \sigma_t \mathbf{z}_1. \tag{10}$$

The model learns to estimate the velocity vector pointing from $\mathbf{x}_0$ to $\mathbf{z}_1$ and the training loss is formulated as:

$$\mathcal{L}_{\text{img-CFM-m}}(\theta) = \mathbb{E}_{t,\mathbf{x}_0,\mathbf{z}_1} \|\mathbf{m} \odot (\hat{\mathbf{v}}_\theta(\mathbf{x}_t, t, \mathbf{c}) - (\mathbf{z}_1 - \mathbf{x}_0))\|_2^2, \tag{11}$$

where $\hat{\mathbf{v}}_\theta$ denotes the model's velocity prediction, $\mathbf{m}$ indicates the masked regions of the latent tokens, and $\mathbf{c}$ comprises $\mathbf{X}_{ctx}$ and $\mathbf{X}_c$. To ensure a concise optimization process, the training excludes additional objectives such as perceptual loss or CTC loss.

### 3.3.3 ROTATED ALIGNED POSITION ENCODING

Directly fine-tuning pretrained diffusion transformer models fails to fully exploit their generative capacity for HTG tasks. We therefore redesign positional encodings to better support multi-line handwritten text with arbitrary line lengths. As shown in Figure 4, naïve 2D RoPE arranges tokens row by row in the concatenated image $\mathbf{I}$, where standard-font and handwritten tokens are interleaved. However, since the length of text-line images varies significantly, the model struggles to distinguish whether a given token should serve as a style condition or a content condition.

To mitigate this issue, we propose Aligned Positional Encoding (APE). In APE, the token arrangement of $\mathbf{I}$ remains unchanged, while the positional encodings assigned to $\mathbf{X}_c$ are directly shared with

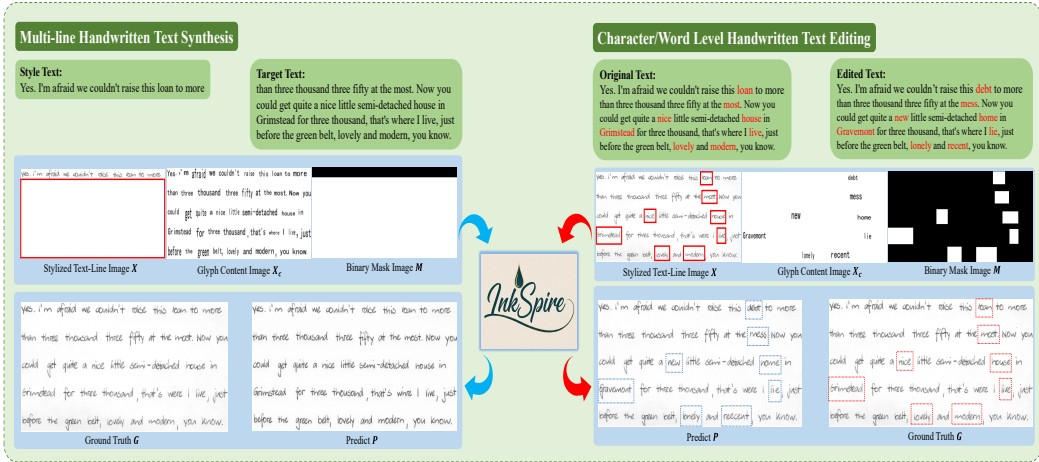

Figure 5: Inference applications of InkSpire, encompassing Multi-line Handwritten Text Synthesis and Character/Word-level Handwritten Text Editing. All applications are guided by the mask image $\mathbf{M}$, the stylized image $\mathbf{X}_s$ and the content image $\mathbf{X}_c$.

their counterparts in $\mathbf{X}$. For cases where text-line images are wider than tall, we further introduce Rotated APE (R-APE): $\mathbf{X}$ and $\mathbf{X}_c$ are rotated 90° clockwise before concatenation, so that target tokens and their content-condition counterparts remain spatially close in the positional space.

## 3.4 APPLICATIONS OF INFERENCE

Owing to the powerful contextual generation ability of diffusion transformer models and the precise spatial control provided by $\mathbf{X}_c$, our framework enables versatile inference through simple adjustments to the mask $\mathbf{M}$ or modifications to $\mathbf{X}_c$. The overall procedure is illustrated in Figure 5.

- Multi-line Handwritten Text Synthesis: Given a single style reference image from a writer, previous approaches are typically constrained to generating only one or a few text lines. In contrast, our model is capable of synthesizing an arbitrary number of text lines simultaneously, by placing the style reference in the first line and masking the remaining parts, conditioned on the multi-line content image $\mathbf{X}_c$.

- Character/Word Level Handwritten Text Editing: By providing a mask that specifies the regions to be edited, together with an edit-content image rendered in standard font, InkSpire can accurately modify multiple words within a handwritten text image while preserving the unmasked regions.

## 4 EXPERIMENTS

## 4.1 EXPERIMENTAL SETTINGS

## 4.1.1 DATASETS

To validate the effectiveness of InkSpire in synthesizing handwritten text-line images, experiments are conducted on the IAM dataset (Marti & Bunke, 2002) for English and on CASIA-HWDB2.0-2.2 (Liu et al., 2011) and ICDAR2013 datasets (Yin et al., 2013) for Chinese. IAM comprises 13,353 English text-line images, with 496 writers' samples for training and the remaining 161 for testing. For Chinese, CASIA-HWDB2.0-2.2 contains 52,230 text-line images from 1,019 writers for training, while ICDAR2013 includes 3432 text-line images from 60 writers for testing. All datasets provide layout annotations at the word level for English and the character level for Chinese.

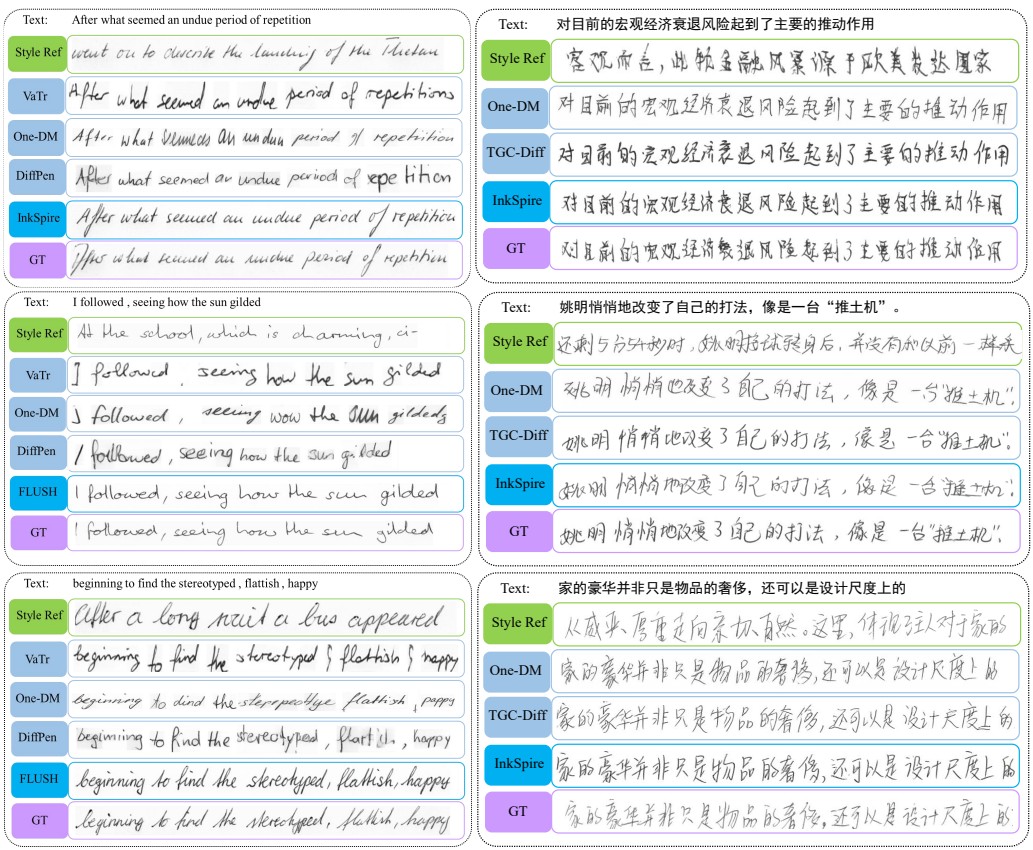

Figure 6: Qualitative comparison between VaTr, One-DM, DiffPen, TGC-Diff, and our proposed InkSpire on image generation across the considered datasets. Our multilingual model demonstrates strong style imitation capabilities under both Chinese and English conditions.

### 4.1.2 EVALUATION METRICS

The evaluation metrics encompass two key aspects: style and content. To assess style diversity and consistency, we employ the Fréchet Inception Distance (FID) (Heusel et al., 2017), Kernel Inception Distance (KID) (Bińkowski et al., 2018), and the task-specific Handwriting Distance (HWD) (Pippi et al., 2023b). For content accuracy, we adopt the Correct Rate (CR) and Accuracy Rate (AR) (Yin et al., 2013) for Chinese text, while for English, we use the Absolute Character Error Rate Difference ($\Delta$CER) (Nikolaidou et al., 2024).

### 4.1.3 COMPARED METHODS

For English handwritten text line generation, we compare our method against representative models, including HWT (Bhunia et al., 2021), VATr (Vanherle et al., 2024), One-DM (Dai et al., 2024), and DiffPen (Nikolaidou et al., 2024). For Chinese handwritten text line generation, the comparison is conducted with One-DM and TGC-Diff (Wang et al., 2025).

### 4.1.4 IMPLEMENTATION DETAILS

For layout generation, we employ 10 transformer layers across all three modeling approaches. During inference, the three layout modeling strategies differ in how layouts are produced: in the autoregressive model, the reference layout is encoded as prefix tokens and the target layout is generated sequentially in a token-by-token manner; in the masked layout model, the input contains an observed reference layout and a masked target layout, and all masked tokens are predicted in a single forward pass; in the Conditional Flow Matching (CFM) model, the same masked positions as in the masked

| Method | IAM Layout | | | | ICDAR2013 Layout | | | |
|---|---|---|---|---|---|---|---|---|
| | $\Delta x\downarrow$ | $\Delta y\downarrow$ | $\Delta w\downarrow$ | $\Delta h\downarrow$ | $\Delta x\downarrow$ | $\Delta y\downarrow$ | $\Delta w\downarrow$ | $\Delta h\downarrow$ |
| **Autoregressive Modeling** | 5.60 | 17.04 | 11.42 | 13.67 | 7.28 | 19.25 | 14.29 | 12.88 |
| **Masked Layout Modeling** | 5.18 | 14.51 | 6.52 | 8.04 | 6.42 | 16.71 | 13.67 | 10.56 |
| **Masked Modeling with CFM** | **4.74** | **14.39** | **4.74** | **4.94** | **5.13** | **14.85** | **12.43** | **8.74** |

Table 1: Layout prediction results on IAM and ICDAR2013 datasets. Values in the table correspond to $\mathcal{L}_1$ losses multiplied by $10^3$; lower values indicate better performance ($\downarrow$).

| | FID$\downarrow$ | KID$\downarrow$ | HWD$\downarrow$ | $\Delta$CER$\downarrow$ |
|---|---|---|---|---|
| **HWT** | 44.72 | 43.49 | 2.97 | 0.33 |
| **VATr** | 34.00 | 29.68 | 2.38 | 0.03 |
| **One-DM** | 43.89 | 44.48 | 2.83 | 0.13 |
| **DiffPen** | 12.89 | 9.73 | 2.13 | 0.03 |
| **InkSpire** | **7.92** | **4.83** | **0.62** | **0.01** |

Table 2: English text line generation results on IAM. The KID is multiplied by $10^3$. Lower is better ($\downarrow$).

| | FID$\downarrow$ | KID$\downarrow$ | HWD$\downarrow$ | CR$\uparrow$ | AR$\uparrow$ |
|---|---|---|---|---|---|
| **One-DM** | 34.36 | 28.37 | 0.80 | 73.19 | 72.33 |
| **TGC-Diff** | 23.43 | 13.85 | 0.63 | 89.99 | 89.13 |
| **InkSpire** | **10.98** | **11.45** | **0.41** | **92.92** | **91.56** |

Table 3: Chinese text line generation results on ICDAR2013. The KID is multiplied by $10^3$. Lower is better ($\downarrow$), higher is better ($\uparrow$).

layout model are used, but the masked tokens are generated through a continuous denoising process solved with a 10-step ODE solver, enabling smooth and flexible layout prediction.

For image generation, our approach builds upon the pre-trained FLUX.1-Fill-dev model Labs (2024), a latent rectified flow transformer tailored for image editing. To adapt to multilingual generation, training samples across languages are integrated within a unified training pipeline. The image patch size is set to $P = 1024$, and the batch size is 4. The model is optimized using the Prodigy optimizer with an initial learning rate of 1 and a weight decay of 0.01. Fine-tuning is performed via Low-Rank Adaptation (LoRA) with a rank of 32 and a LoRA scaling factor of 32, trained for 20,000 iterations on four A100 GPUs (40 GB). In total, the LoRA modules introduce approximately 115.9M trainable parameters. A comprehensive description of all fine-tuned LoRA parameters can be found in Appendix Table 8. The fine-tuned model achieves high-quality synthesis with about 20 ODE denoising steps. For more implementation details of layout generation and image generation, please refer to Appendix A.3.

## 4.2 ADDITIONAL ANALYSIS OF STYLIZED LAYOUT GENERATION

We compare three layout modeling strategies—Autoregressive, Masked, and Masked with Conditional Flow Matching (CFM)—using average losses of four layout-specific features. For all methods, the tokens corresponding to the first line of each paragraph are provided as reference tokens, and the models are tasked with predicting the layout parameters of all remaining tokens. Under this setting, the masked layout model and CFM apply masking to all non-reference tokens. The evaluation metric is computed by averaging the L1 deviations over the full set of tokens that require prediction. As shown in Table 1, CFM consistently outperforms the others, demonstrating its effectiveness in capturing complex spatial dependencies while preserving layout coherence.

## 4.3 STYLIZED HANDWRITTEN TEXT-LINE GENERATION

Although InkSpire is trained on multi-line images to capture in-context stylistic features, we follow the one-shot evaluation protocol, where only a single reference text-line is provided. The goal is to generate new lines with the same style but different content. During inference, we consistently select references by using the second line to generate the first, and then the first line for the remaining ones, which mitigates intra-writer style variations across paragraphs.

Quantitative results, as presented in Table 2 and Table 3, demonstrate that InkSpire consistently outperforms state-of-the-art methods on both English and Chinese datasets. As illustrated in Figure 6, our model not only produces handwritten styles that are more visually aligned with the reference,

|          | FID↓  | KID↓  | HWD↓ | ΔCER↓ |
|----------|-------|-------|------|-------|
| baseline | 15.12 | 19.27 | 0.97 | 0.11  |
| +APE     | 9.31  | 7.21  | **0.58** | 0.05  |
| +R-APE   | **7.92** | **4.83** | 0.62 | **0.01** |

Table 4: Ablation study of positional encoding on the IAM Lines dataset. The KID is multiplied by $10^3$. Lower is better (↓).

|          | FID↓  | KID↓  | HWD↓ | CR↑   | AR↑   |
|----------|-------|-------|------|-------|-------|
| baseline | 17.57 | 18.61 | 0.53 | 89.72 | 88.91 |
| +APE     | 11.75 | 12.35 | 0.42 | 91.87 | 90.63 |
| +R-APE   | **10.98** | **11.45** | **0.41** | **92.92** | **91.56** |

Table 5: Ablation study of positional encoding on the ICDAR2013 Lines dataset. The KID is multiplied by $10^3$. Lower is better (↓), higher is better (↑).

|           | FID↓  | KID↓  | HWD↓ | ΔCER↓ |
|-----------|-------|-------|------|-------|
| F-TopMask | 8.73  | 6.13  | 0.78 | 0.07  |
| R-Mask    | **7.92** | **4.83** | **0.62** | **0.01** |

Table 6: Ablation study of masking strategieson the IAM Lines dataset. The KID is multiplied by $10^3$. Lower is better (↓).

|           | FID↓  | KID↓  | HWD↓ | CR↑   | AR↑   |
|-----------|-------|-------|------|-------|-------|
| F-TopMask | 11.57 | 13.41 | 0.48 | 92.48 | 91.34 |
| R-Mask    | **10.98** | **11.45** | **0.41** | **92.92** | **91.56** |

Table 7: Ablation study of masking strategies on the ICDAR2013 Lines dataset. The KID is multiplied by $10^3$. Lower is better (↓), higher is better (↑).

but also exhibits significantly improved character structure accuracy. These findings provide strong evidence of the superiority of our approach and are well-aligned with the quantitative metrics.

## 4.4 ABLATION STUDY

### 4.4.1 ABLATION STUDY ON POSITION ENCODING

From Table 4 and Table 5, we observe that Aligned Positional Encoding (APE) markedly enhances style diversity and content accuracy by helping the model better distinguish style from content tokens. While naïve positional encoding can handle single-line generation, it often copies input images in multi-line settings and is sensitive to resolution. Moreover, the rotated variant (R-APE) yields further gains on both English and Chinese datasets, as it better localizes tokens in long text lines under the one-shot setting.

### 4.4.2 ABLATION STUDY ON MULTI-LINE MASKED INFILLING STRATEGY

In addition to the Random-Size Multi-Region Masking (R-Mask) strategy described in Section 3.3.1, we introduce a Fixed Top-Region Unmasked Masking (F-TopMask) scheme as an ablation variant. Since masking is performed on a 1024×1024 image patch, the fixed-mask design keeps only the top 128×1024 region visible while masking all remaining lower areas. This setup resembles the inference scenario where only the first text line is provided as a style reference and the model must generate all subsequent lines. The quantitative results in Table 6 and Table 7 demonstrate that our R-Mask strategy yields moderate improvements over the fixed-region alternative.

## 5 CONCLUSION

We introduced InkSpire, a diffusion transformer that unifies style, content, and noise for handwritten text generation. By removing explicit encoders and adopting multi-line masked infilling with revised positional encoding, it enables efficient training, arbitrary-length synthesis, and fine-grained editing. Trained on English and Chinese corpora, InkSpire outperforms prior methods in fidelity and stylistic diversity. Future work will extend to more languages and datasets to enhance generalization.

**Acknowledgment:** This work is supported by the Natural Science Foundation of China (NSFC) Grant No. 62276258 and No. U23B2029.

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

# A APPENDIX

## A.1 INFORMATION ABOUT USE OF AI ASSISTANTS

The authors used AI-assisted tools solely for language polishing, including grammar, spelling, and expression refinement. All conceptual contributions, technical innovations, experimental design, and analyses presented in this work were independently performed by the authors without reliance on AI for content generation or scientific reasoning.

## A.2 USER STUDIES

### A.2.1 USER PREFERENCE STUDY

We conduct a human evaluation to assess the perceptual quality of synthesized Chinese handwritten text-line images, focusing on style fidelity and content correctness. Participants, all of whom hold postgraduate-level education, compare the outputs of our method with two state-of-the-art baselines, One-DM and TGC-Diff. In each trial, a writer is randomly sampled from the ICDAR2013 dataset, and one of their handwritten text-line images is provided as a style reference, together with an identical content prompt for all methods. Participants are shown the reference text-line alongside multiple candidate images generated by the three models and are asked to select the sample with the highest overall generation quality. Figure 7 illustrates an example of the questionnaire instructions and its corresponding question items. The evaluation consists of 30 rounds, yielding 900 valid responses from 30 volunteers. As shown in the Figure 8(a), our method receives the highest number of user selections, indicating its superior perceptual quality in handwritten text generation.

**Instructions**
This survey contains **30 evaluation questions**.
For each question, you will be given **a target text** and **a style reference image**.
Options **A, B, and C** show images generated by **three different models** (in randomized order).
Your task is to **select the image with the highest generation quality**.

**"Generation quality"** primarily includes two aspects:
**1. Style:** How similar the generated handwriting style is to the reference
(e.g., stroke thickness, curvature, connections, and other local stylistic details).
**2. Content:** The correctness and clarity of the generated characters.
**Note:** If the reference writing style is exaggerated, easily distorted, or blurry, please prioritize style consistency over content accuracy. Please ignore occasional hallucinated content that exceeds the given target text.

In short, choose the image that **most closely resembles being written by the same person** as in the style reference image.For best results, please complete the survey **on a computer** and **click to zoom in** on each image before making a selection.

要生成的文本内容

退风险起到了主要的推动作用,且其本身仅仅是全球宏观

风格参考图片

客观而言,此抢金融风暴源于欧美发达国家作用建于发监

A.

退风险起到了主要的推动作用,且其本身仅仅是全球宏观

B.

退风险起到了主要的推动作用,且其本身仅仅是全球宏观

C.

退风险起到了主要的推动作用,且其本身仅仅是全球宏观

Figure 7: User Preference Study Instructions.

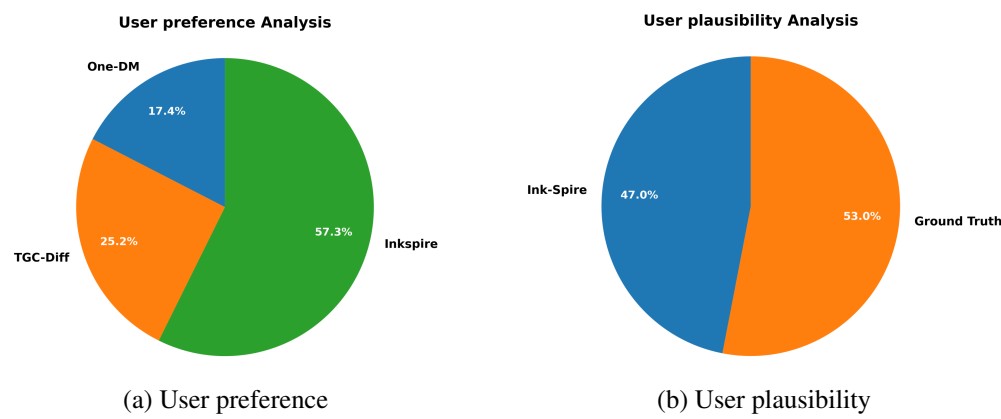

(a) User preference          (b) User plausibility

Figure 8: Overall caption describing both images.

### A.2.2 USER PLAUSIBILITY STUDY

We conduct a user plausibility study to evaluate whether text-line images generated by InkSpire are perceptually indistinguishable from real handwriting. Participants are first presented with 30 authentic handwritten text-line samples, which serve as style reference images. In each question, they are then shown two candidate images: one genuine sample written by the same author and one generated by our model. Their task is to determine which image appears more likely to be written by the same writer as the reference. Figure 9 illustrates an example of the questionnaire instructions and its corresponding question items. A total of 23 participants provide 690 valid responses. As shown in Figure 8(b), the selection accuracy converges to approximately, indicating performance at chance level. This suggests that the text-line images produced by InkSpire are nearly indistinguishable from real handwriting.

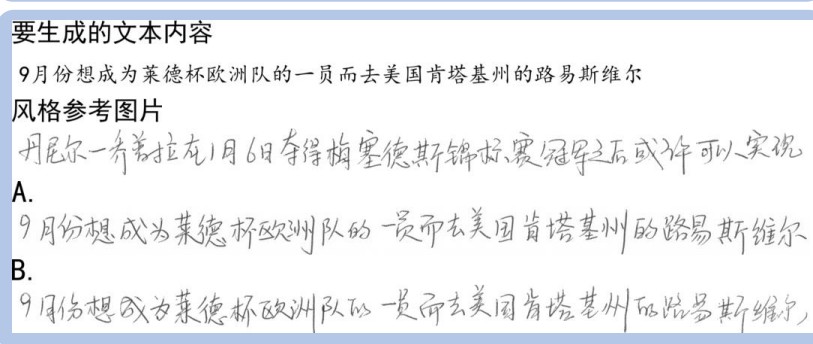

Figure 9: User Plausibility Study Instructions.

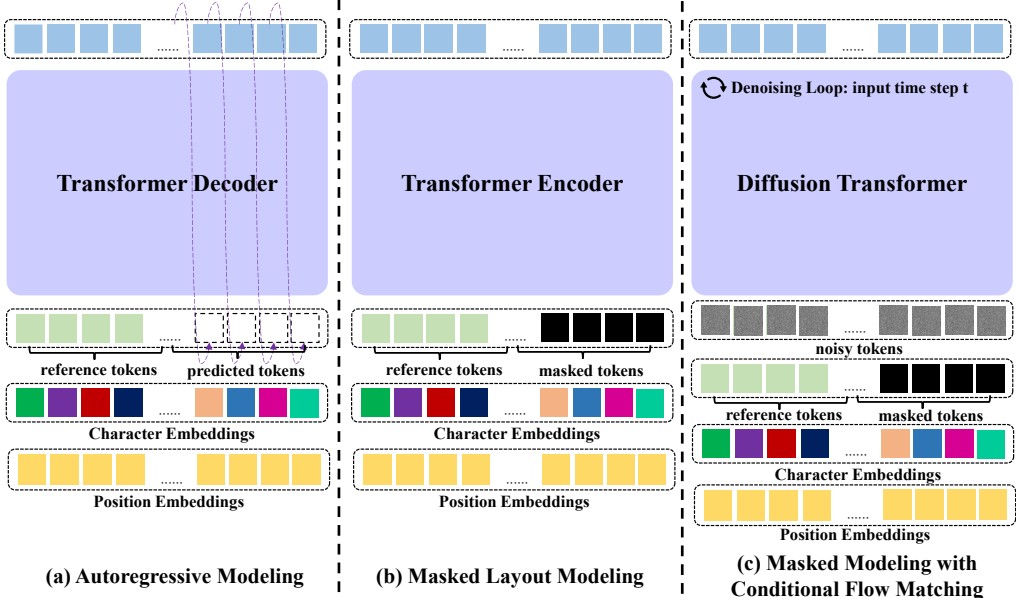

Figure 10: Layout Inference Architectures.

### A.3 MORE IMPLEMENTATION DETAILS

#### A.3.1 LAYOUT GENERATION DETAILS

We provide a detailed description of the three layout generation strategies discussed in the main text:

- Autoregressive Modeling: This approach predicts the layout of each target token using an autoregressive Transformer architecture. The model input consists of the embedding of the current character (or, for English words, the sum of the individual character embeddings to form a word-level embedding), positional embeddings, and the layout information of preceding tokens. We employ 10 Transformer decoder layers and optimize the model using an L1 regression loss.

- Masked Layout Modeling: In this strategy, the input consists of character embeddings, positional embeddings, and a partially masked layout representation of tokens. We adopt two masking strategies: (1) randomly masking a contiguous segment of arbitrary length, and (2) masking each token independently with a 20% probability. Each masking strategy is applied with equal probability (50%). The model is trained to predict the layout of the masked tokens. We employ 10 Transformer encoder layers and optimize the model using an L1 regression loss computed over the masked tokens.

- Masked Modeling with Conditional Flow Matching: This variant adopts the same masking strategies as Masked Layout Modeling, but extends the masked modeling framework by incorporating a time-dependent denoising condition inspired by flow-matching objectives. In addition to character embeddings, positional embeddings, and conditional tokens, all masked tokens are progressively denoised from noisy tokens over multiple timesteps. We employ a 10-layer diffusion Transformer to model this process and optimize it using an L1 loss over the predicted layout values (v-prediction) for the masked tokens.

During inference, we take the tokens of the first sentence in a paragraph as reference tokens and ask the models to predict the remaining tokens. The detailed model structures are illustrated in Figure 10. All Transformer blocks have a hidden dimension of 512 and employ 8 attention heads. The models are trained with a batch size of 160 using the AdamW optimizer with a base learning rate of $1 \times 10^{-4}$.

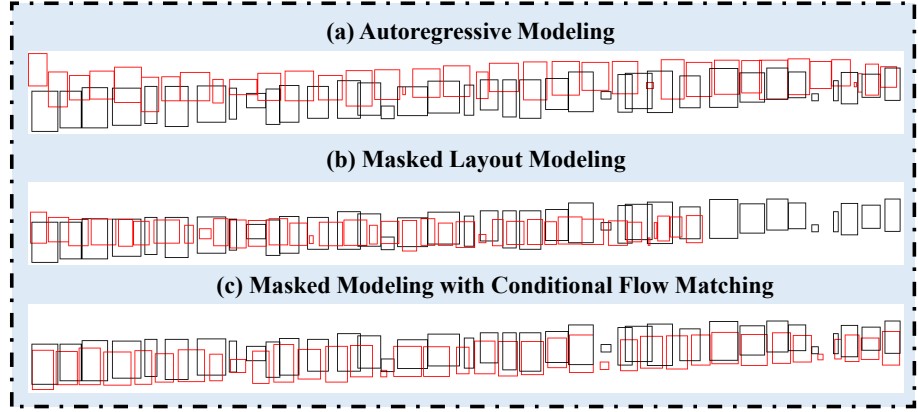

Figure 11: Visualization results of Chinese text-line generation. Black rectangles indicate the target layout, while red rectangles denote the predicted bounding boxes.

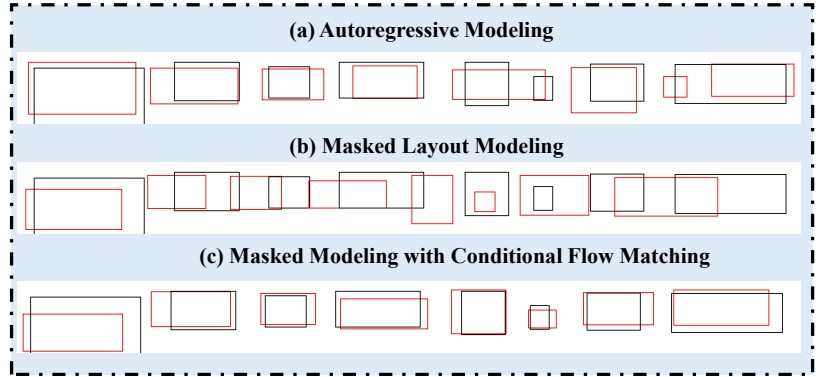

Figure 12: Visualization results of English text-line generation. Black rectangles indicate the target layout, while red rectangles denote the predicted bounding boxes.

### A.3.2 HANDWRITTEN TEXT IMAGE GENERATION DETAILS

We list all LoRA fine-tuning parameters in Table 8. With a LoRA rank of 32 and a scaling factor of 32, the LoRA modules collectively introduce approximately 115.9M trainable parameters.

### A.4 ADDITIONAL ANALYSIS OF STYLIZED LAYOUT GENERATION

We provide two visualization examples for Chinese and English layout generation in Figure 11 and Figure 12, respectively. In each figure, black rectangles denote the target layout, while red rectangles represent the predicted character bounding boxes produced by each method.

In Figure 11, the Chinese example illustrates that our masked modeling with CFM successfully captures the upward-slanting trajectory of the text line—from the lower left toward the upper right. In contrast, the slant is much less pronounced in the layouts generated by the Autoregressive Model and the Masked Layout Modeling baseline.

In Figure 12, the English example shows that masked modeling with CFM provides noticeably better control over word spacing and word size compared with the other two strategies, leading to a more coherent and visually consistent layout.

Overall, the layouts generated by masked modeling with CFM demonstrate superior visual fidelity, aligning well with the quantitative improvements reported in Table 1.

| Parameter Name | #Params (M) |
|---|---|
| x_embedder | 0.1106 |
| transformer_blocks.0–18.norm1.linear | 13.072 |
| transformer_blocks.0–18.attn.to_q | 3.7355 |
| transformer_blocks.0–18.attn.to_k | 3.7355 |
| transformer_blocks.0–18.attn.to_v | 3.7355 |
| transformer_blocks.0–18.attn.to_out.0 | 3.7355 |
| transformer_blocks.0–18.ff.net.2 | 9.3389 |
| single_transformer_blocks.0–37.norm.linear | 14.942 |
| single_transformer_blocks.0–37.proj_mlp | 18.678 |
| single_transformer_blocks.0–37.proj_out | 22.413 |
| single_transformer_blocks.0–37.attn.to_q | 7.4711 |
| single_transformer_blocks.0–37.attn.to_k | 7.4711 |
| single_transformer_blocks.0–37.attn.to_v | 7.4711 |
| **Total** | **115.91** |

Table 8: List of all LoRA parameter names.

| Model | Style Score (%) |
|---|---|
| **English Models** | |
| HWT | 39.62 |
| VATr | 45.19 |
| One-DM | 50.73 |
| DiffPen | 61.26 |
| InkSpire | **78.58** |
| **Chinese Models** | |
| One-DM | 51.34 |
| TGC-Diff | 60.46 |
| InkSpire | **86.28** |

Table 9: Style Scores (%) of different models for English and Chinese text-line generation.

## A.5 ADDITIONAL ANALYSIS OF STYLIZED HANDWRITTEN TEXT-LINE GENERATION

### A.5.1 VISUALIZATION ANALYSIS OF POSITION ENCODING ABLATION STUDY

We provide visualization results for the ablation study on positional encoding. Figure 13 presents examples of both Chinese and English text-line generation. The red bounding boxes highlight instances where the model produces structurally incorrect handwriting.

From the visualizations, we observe that using the original positional encoding of the Flux-fill backbone leads to inaccurate spatial localization, causing the model to generate handwriting fragments at incorrect positions within the image. In contrast, APE improves positional consistency, and the proposed R-APE further enhances structural accuracy. The superiority of R-APE over APE is particularly evident in the correctness of character shapes and alignment, which is consistent with the quantitative improvements reported in Table 4 and Table 5.

### A.5.2 VISUALIZATION ANALYSIS OF MASKED INFILLING STRATEGY ABLATION STUDY

We provide visualization results of the ablation study on masked infilling strategy. Figure 14 presents examples of both Chinese and English text-line generation. In both figures, the red bounding boxes highlight cases where the model produces structurally incorrect handwriting.

Overall, the stylistic appearance produced by both the random masking (R-Mask) and the fixed top-region masking (F-TopMask) strategies remains comparable and closely aligned with the target style. However, subtle differences emerge at the fine-grained character-structure level: the R-Mask strategy leads to fewer structural errors, likely because its randomly sampled masks expose the model to a wider range of local character patterns during training. These qualitative observations are consistent with the quantitative improvements reported in Table 6 and Table 7.

### A.5.3 STYLE SCORE EVALUATION

We evaluate style fidelity using text-line style classifiers built upon ImageNet-pretrained Swin Transformers (about 86.8M parameters), trained separately for English and Chinese. All text-line images are normalized to 64×1024 resolution by proportional resizing to height 64 followed by width-wise cropping or white padding.

The resulting classifiers reach 94% accuracy on the 60-way Chinese validation set and 91% on the 161-way English set. Their prediction accuracy on generated text lines is reported as the *style score*, indicating how closely a model replicates the target handwriting style.

As shown in Table 9, InkSpire consistently attains the highest style scores across both English and Chinese settings, outperforming all baselines and aligning with trends observed in the user study and earlier style-consistency analyses.

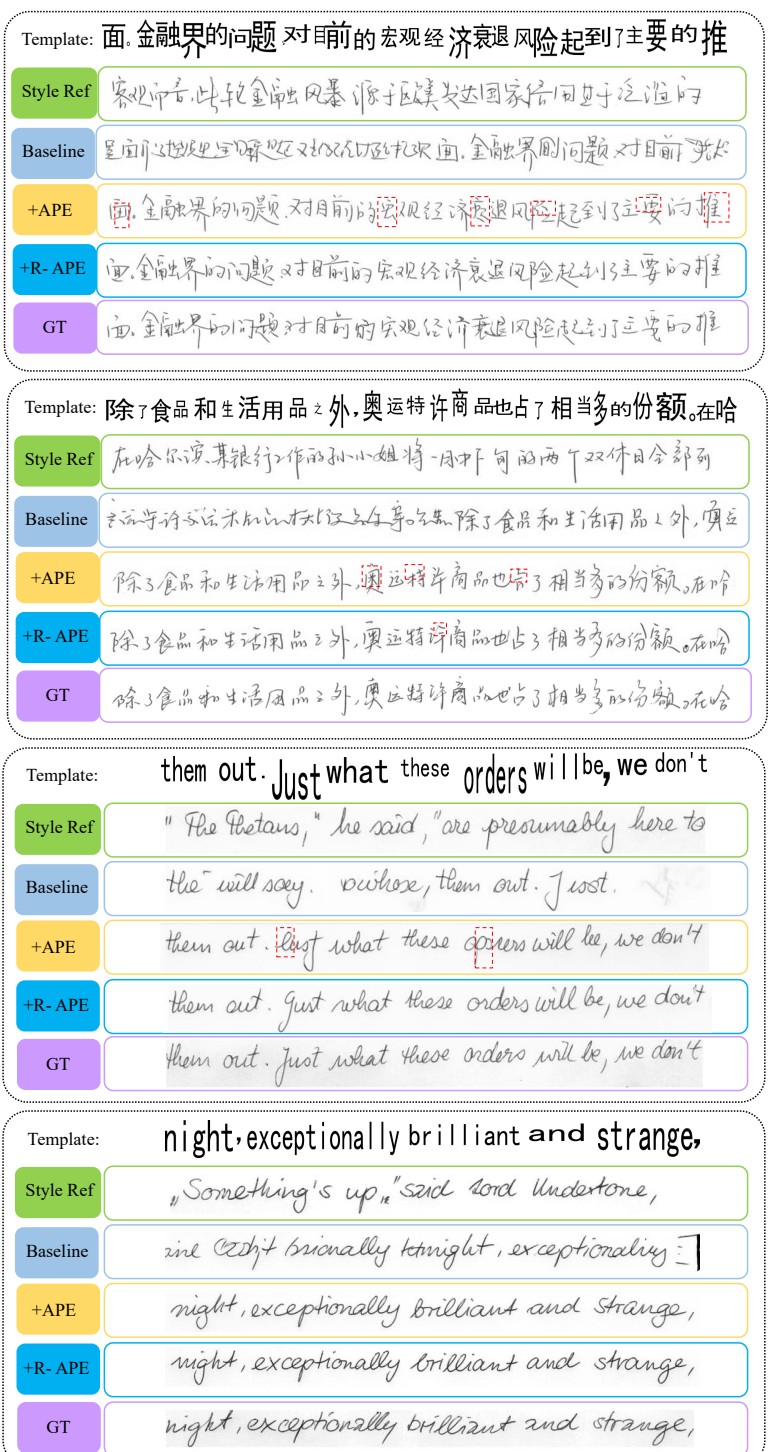

Figure 13: Visualization results of Chinese text-line generation for the positional encoding ablation. Red rectangles highlight cases where incorrect character structures are produced.

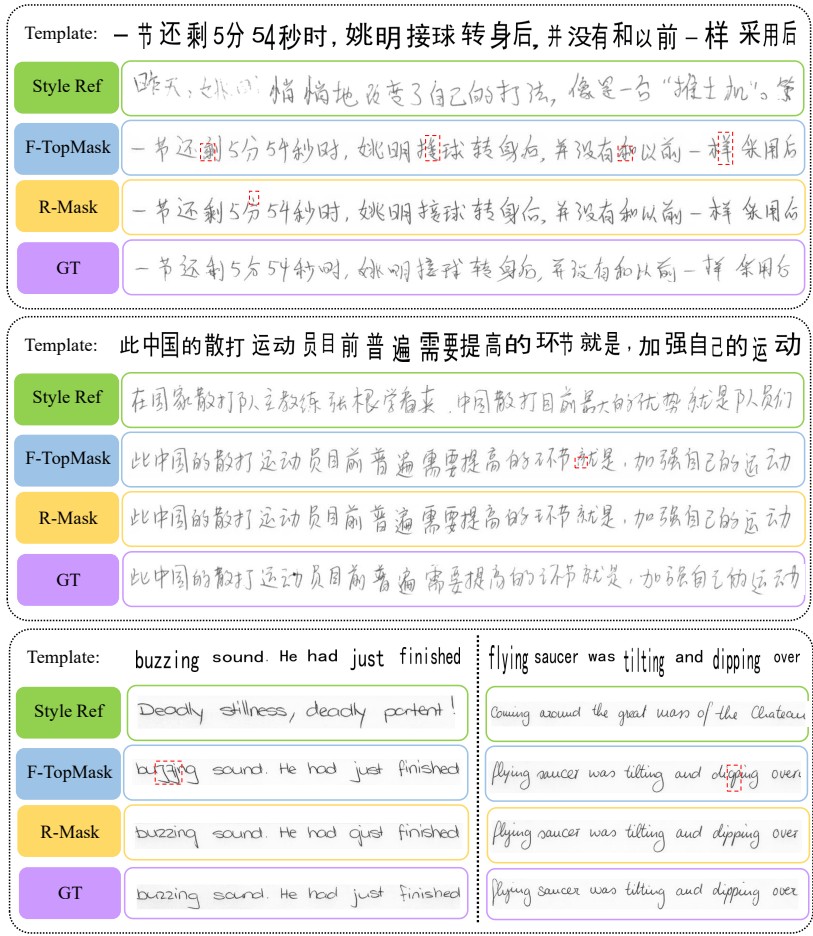

Figure 14: Visualization results of Chinese and English text-line generation for the masked infilling strategy ablation. Red rectangles highlight cases where incorrect character structures are produced.

### A.5.4    ANALYSIS OF INTER-LINE STYLE CONSISTENCY

We define inter-line style consistency as the stylistic coherence between multiple generated text lines that share the same textual content but use different style reference lines from the same handwriting passage. As discussed in Section 3.3.1, compressing reference lines with different slant angles to a fixed height introduces mismatched character scales, which leads to noticeable style discrepancies among the generated lines. Moreover, the reduced resolution further weakens the model's ability to capture fine-grained stylistic cues.

As shown in Figure 15, when different style-reference lines are used to condition the generation of the same text, visible stylistic variations emerge. Highly slanted reference lines, once height-normalized, become excessively small, making it difficult for the model to extract meaningful style information and resulting in outputs resembling printed fonts. In contrast, when the original-size reference is used to generate multiple lines simultaneously, the resulting handwriting demonstrates substantially improved inter-line style coherence. A similar phenomenon is observed for English text-line generation, as illustrated in Figure 16.

In summary, maintaining the native resolution in training and inference markedly strengthens the model's stylistic imitation capability and inter-line style consistency, while reducing the style instability caused by slope-induced compression artifacts.

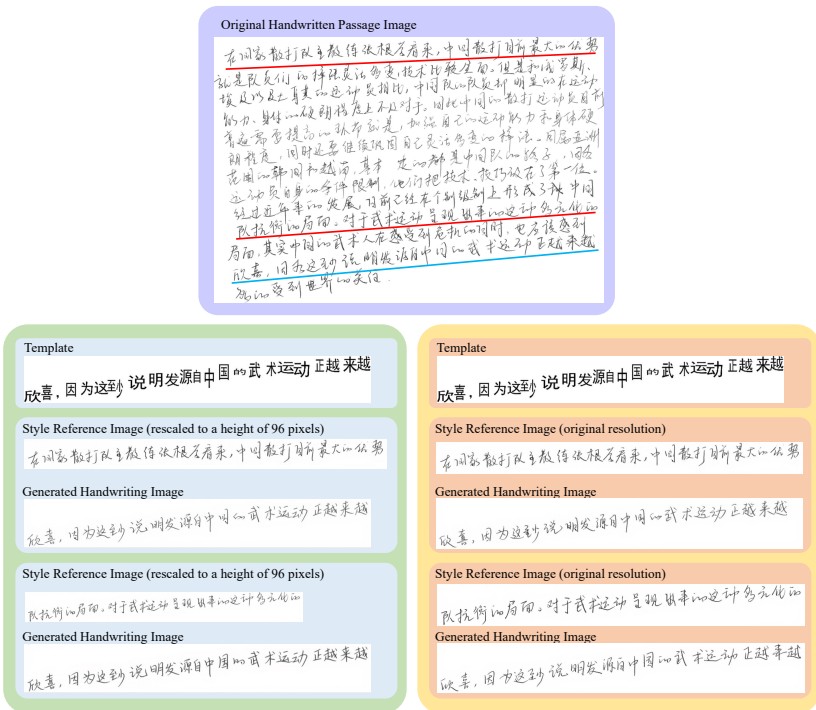

Figure 15: Visualization results of Chinese text-line generation for evaluating inter-line style consistency. Red lines indicate the style-reference text line, and blue lines indicate the target text line.

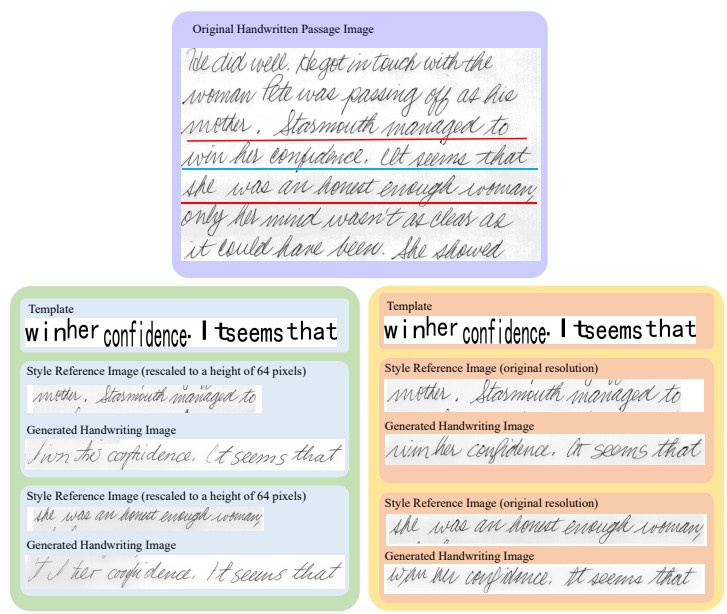

Figure 16: Visualization results of English text-line generation for evaluating inter-line style consistency. Red lines indicate the style-reference text line, and blue lines indicate the target text line.

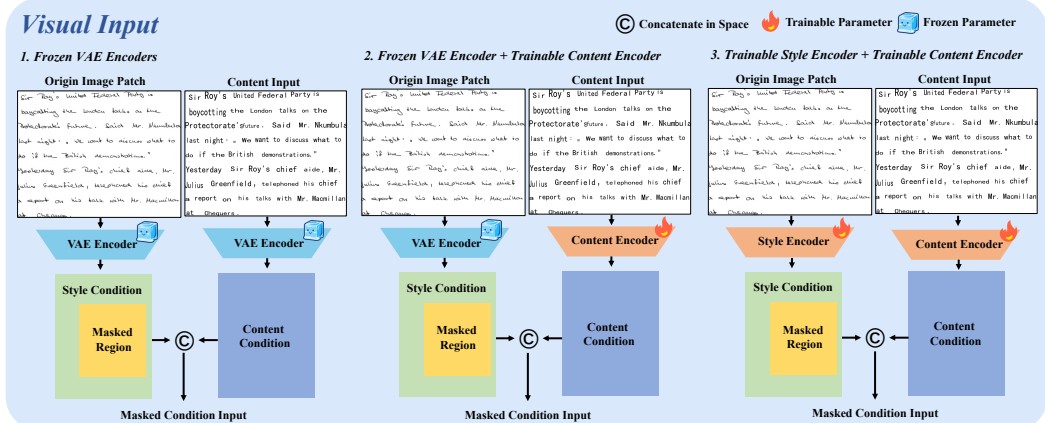

Figure 17: Overview of the visual input processing under different encoder configurations.

|            | FID↓  | KID↓  | HWD↓ | ΔCER↓ |
|------------|-------|-------|------|-------|
| **Fixed-VAE** | **7.92** | **4.83** | **0.62** | **0.01** |
| C-Enc      | 20.48 | 10.31 | 1.58 | 0.05  |
| C+S-Enc    | 17.39 | 8.97  | 0.71 | 0.02  |

Table 10: Ablation of encoder configurations on the IAM Lines dataset. C-Enc: only Content Encoder trained from scratch; C+S-Enc: both Content and Style Encoders trained from scratch. KID is scaled by $10^3$.

|            | FID↓  | KID↓  | HWD↓ | CR↑   | AR↑   |
|------------|-------|-------|------|-------|-------|
| **Fixed-VAE** | **10.98** | **11.45** | **0.41** | 92.92 | 91.56 |
| C-Enc      | 17.46 | 19.52 | 0.91 | 59.32 | 55.58 |
| C+S-Enc    | 16.53 | 14.37 | 0.59 | **94.74** | **93.48** |

Table 11: Ablation of encoder configurations on the ICDAR2013 Lines dataset. C-Enc: only Content Encoder trained from scratch; C+S-Enc: both Content and Style Encoders trained from scratch. KID is scaled by $10^3$.

### A.5.5 IMPACT OF ENCODER DESIGN CHOICES

We conduct an ablation study to analyze the impact of different encoder configurations in our unified encoder-less framework. As illustrated in Figure 17, we design three experimental settings:

- **Pretrained-VAE Encoding (Default Setting).** In the original setup, a pretrained VAE is used to encode both the origin image patch and the content image.
- **Learned Content Encoder Only (C-Enc).** In this variant, we train a content encoder from scratch to extract content features, while the origin image patch continues to be encoded using the frozen pretrained VAE.
- **Jointly Learned Content and Style Encoders (C+S-Enc).** In the third configuration, both the content encoder and the style encoder are jointly learned from scratch.

Following Dai et al. (2024), both the content and style encoders adopt a CNN+Transformer architecture. The CNN backbone is a ResNet-50 pretrained on ImageNet, followed by a Transformer encoder with 3 layers, a hidden dimension of 2048, and 16 attention heads. To adapt the representation to the DiT input format, we append a linear projection layer that maps 2048-dimensional features to 64 dimensions. The proposed encoder contains approximately 124.4M parameters, while the frozen VAE encoder has about 34.3M parameters. Quantitative results for the three configurations are reported in Table 10 and Table 11.

Figure 18 presents visualization results for Chinese and English text-lines. We observe that when training encoders from scratch (both C-Enc and C+S-Enc), the generated strokes tend to appear noticeably lighter. This is likely caused by a distributional bias between the latent space learned from scratch and the latent space of the pretrained VAE. Furthermore, for the C-Enc setting, the model more easily captures complex cursive styles but struggles to maintain accurate character structures. In contrast, the C+S-Enc setting produces outputs that resemble cleaner, more printed-like handwriting, but with reduced stylistic diversity. This suggests that jointly learning content and style encoders may shift the model toward a different balance between content fidelity and stylistic variation. No-

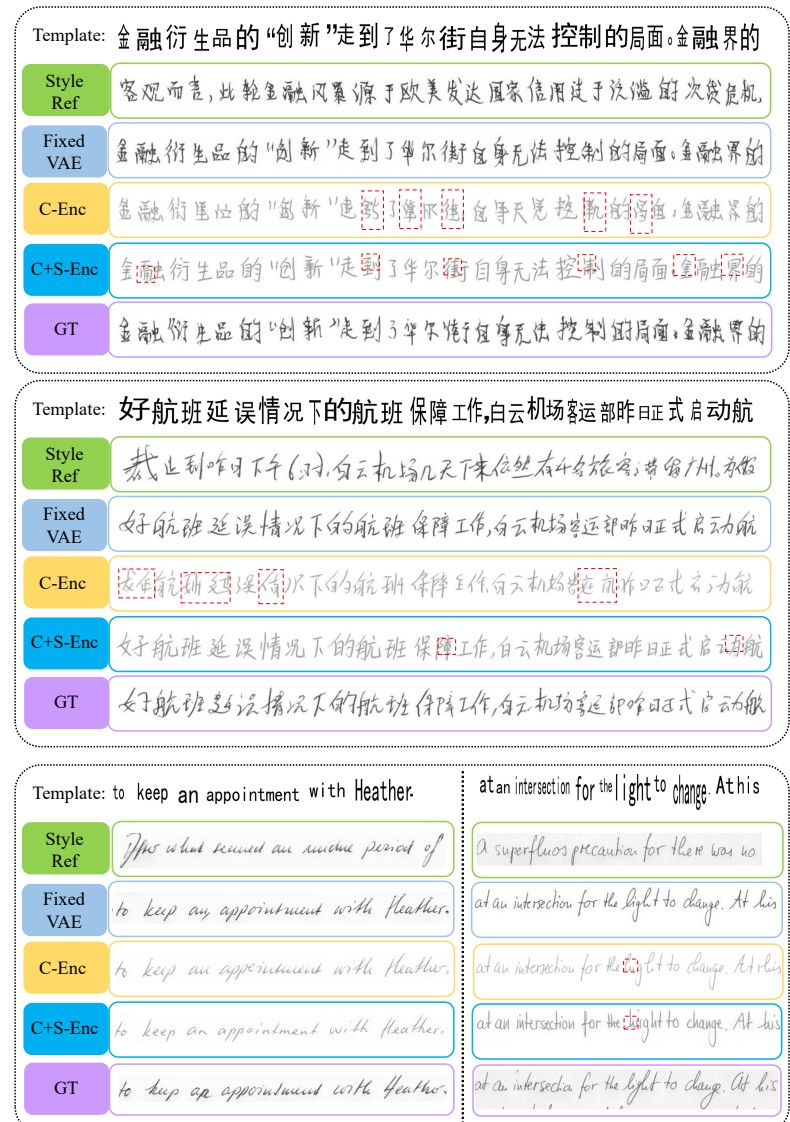

Figure 18: Handwritten Chinese and English text-line generation results under different encoder configurations. Red rectangles indicate cases where incorrect character structures are produced.

tably, Table 10 shows that C+S-Enc achieves higher CR and AR scores than the frozen-VAE-based setup. This is because although C+S-Enc often generates characters with multiple extra strokes, these artifacts rarely mislead the downstream text-line recognition model, resulting in higher recognition metrics despite localized structural errors.

Compared with Chinese, English exhibits fewer character-level errors and generally acceptable styles, although the overall faint-stroke phenomenon still exists. This further indicates that Chinese handwriting is more challenging to model than English. Similarly, the C+S-Enc configuration yields more neatly shaped English words.

Overall, for both Chinese and English, the C-Enc and C+S-Enc settings underperform the frozen-VAE baseline. This may be partly attributed to the inherent alignment between the pretrained VAE's latent space and the subsequent Diffusion Transformer, which provides a stronger and more stable representation than training the encoders from scratch.

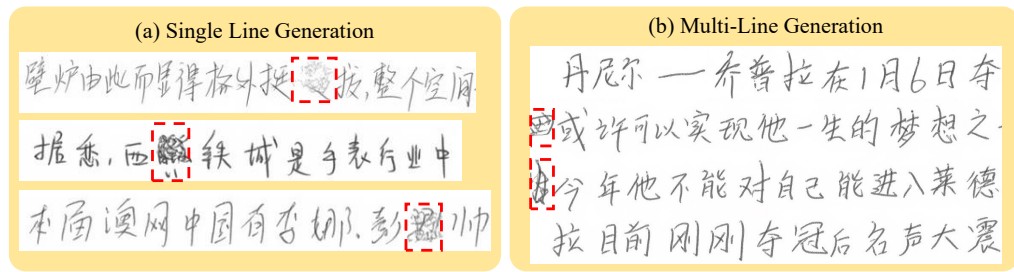

Figure 19: Visualization results of failure-case analysis. The red boxes highlight the erased marks mistakenly generated by the model.

### A.5.6 ANALYSIS OF FAILURE CASES

A common failure mode we observed is that the model sometimes fills in the spaces of a text line with artifacts that resemble crossed-out or erased characters. This behavior stems from the training data: some authors habitually strike out a mistaken character and then write the correction beside it. When the generation is strictly conditioned by the content template, the correct action would be to preserve an empty region (i.e., a gap) for the removed character; instead, the model often produces residual "erasure" marks in that region. The problem is particularly pronounced at paragraph-level generation and near image boundaries. Representative examples are shown in Figure 19.

### A.5.7 ANALYSIS OF STYLE–CONTENT CONTROLLABILITY AND DISENTANGLEMENT

We conduct an explicit style–content disentanglement experiment to verify that InkSpire maintains independent controllability over style and content, despite embedding style, content, and noise into a unified latent space. Given a fixed content template, we generate handwriting samples conditioned on multiple distinct style-reference lines sourced from different writers. This setup evaluates whether the model can vary stylistic attributes (e.g., stroke width, slant angle, character curvature) while keeping the textual content unchanged.

As shown in the visualization results (Figures 20):

- Pink boxes highlight the handwritten style of the letter g, and purple boxes highlight the style of the character sequence "th". We observe that within each sample (e.g., sample a or b), all occurrences of g and th exhibit highly consistent writing patterns that closely match their respective style-reference samples. Meanwhile, samples a and b show strong stylistic divergence from each other, reflecting the differences in their reference styles.

- The red arrows indicate the slant direction of characters in samples c and d. Sample c shows a uniformly left-leaning slant across almost all words, matching its reference style, while sample d shows a consistently right-leaning slant at nearly uniform angles—again mirroring its style reference.

These observations confirm that the unified latent representation does not collapse style and content. Instead, InkSpire successfully preserves the content while expressing distinct stylistic attributes, demonstrating robust and independent controllability over both factors.

### A.6 MORE VISUALIZATION EXAMPLES FOR MULTI-LINE TEXT GENERATION AND EDITING

We provide additional visualization results of InkSpire for multi-line handwritten text generation and character-level handwritten text editing. Figure 21 presents examples of English multi-line text generation, while Figure 22 shows the corresponding results for Chinese. Moreover, Figure 23 illustrates character-level editing results on English handwritten passages, and Figure 24 displays the editing results on Chinese passages.

Figure 20: Visualization results of Style–Content Controllability and Disentanglement. Pink boxes mark stylistic variations of "g", purple boxes highlight "th", and red arrows indicate the global slant direction.

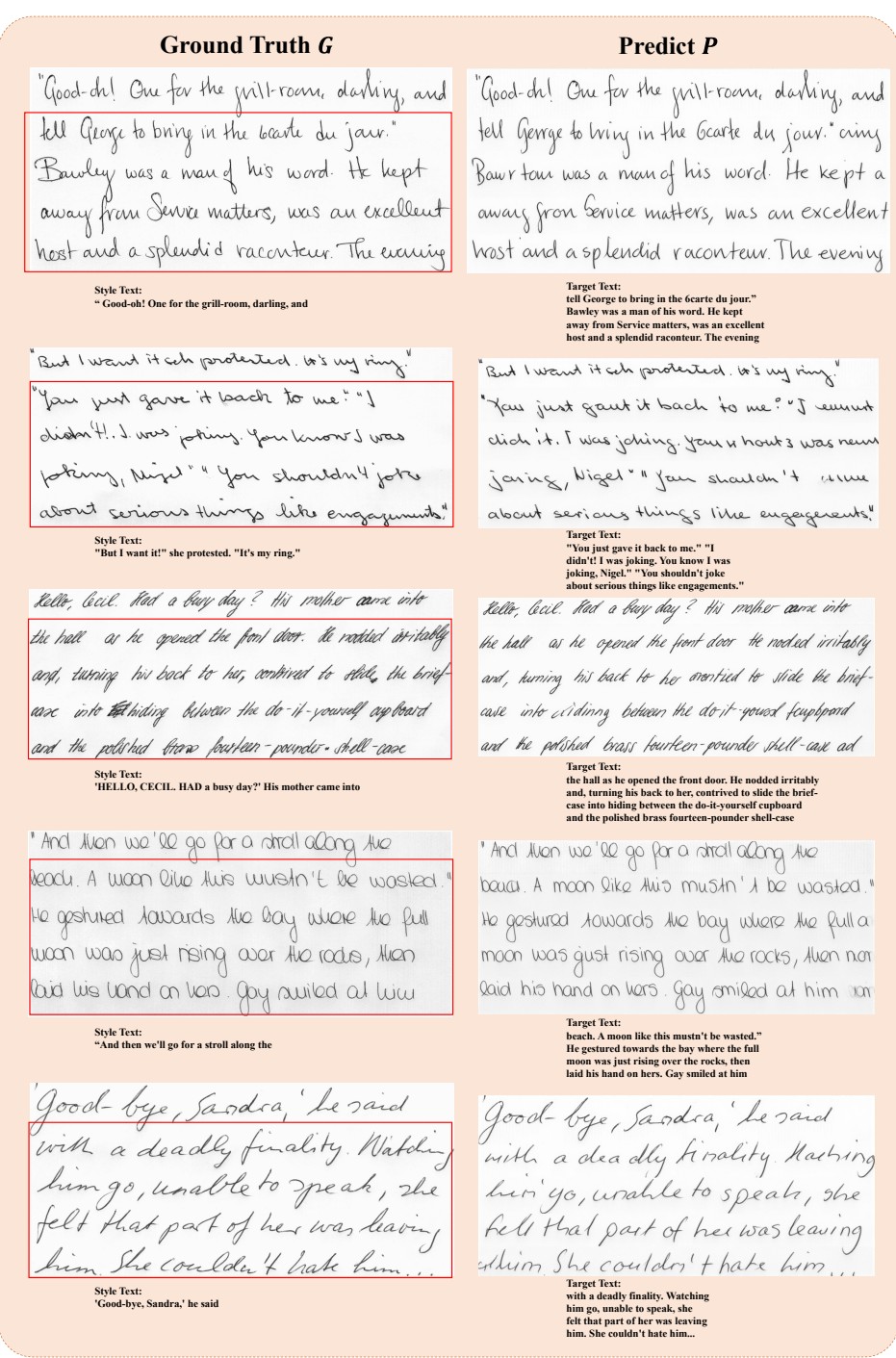

Figure 21: More examples of multi-line English paragraph generation.

**Ground Truth *G***  **Predict *P***

Style Text:
客观而言,此轮金融风暴源于欧美发达国家信用过于泛滥

Target Text:
金融衍生品的"创新"走到了华尔街自身无法控制的局面
问题,对目前的宏观经济衰退风险起到了主要的推动作身
仅仅是全球宏观经济周期的一个表征。按照一般的
理论,任何以市场为主体的经济运行都将经历周期性的

Style Text:
截止到昨日下午6时,白云机场几天下来依然有近千名旅

Target Text:
滞留广州。为做好航班延误情况下的航班保障工作,白场
客运部昨日正式启动航班不正常预案:客运部临时
立了航班不正常指挥小组,组建了由值机员、服务
任的航班不正常处理小组,专人负责航班延误信息的

Style Text:
本届澳网中国有李娜、彭帅、晏紫、袁梦四人参赛,其中三人首轮便

Target Text:
淘汰,李娜在第三轮被世界排名第146位的波兰选手多马乔斯卡击
李娜尽管拥有了亚洲一姐的实力,但在欧美高手林立的女单项目中,李
尚不具备夺取冠军的实力。彭帅一度被寄予厚望,就连华裔网球明
德培都对其很看好,并在去年主动提出担任彭帅的教练。然而,彭帅

Style Text:
沪苏浙高速公路江苏段日前正式建成通车,全线采用六车道标准

Target Text:
车速为120公里/小时,全线有桥梁47座,互通立交7处。沪苏浙高
路江苏段是国家高速公路网中上海至重庆高速公路网的重要组成部
据江苏省高速公路指挥部相关负责人介绍,沪苏浙高速公路江苏段
江苏省的最南端,起点位于吴江市芦墟镇北的苏沪两省市交界

Style Text:
联邦公开市场委员会(FOMC)表示:"虽然短期融资市

Target Text:
压力稍微得到减轻,但金融市场情况却继续恶化;对于一
业以及个人来说,贷款变得更为困难。而且,新的信息反映
市场更加低迷,而劳工市场也出现疲软迹象。"其发布的
还称:可预见的增长减缓的风险继续存在。美联储将

Figure 22: More examples of multi-line Chinese paragraph generation.

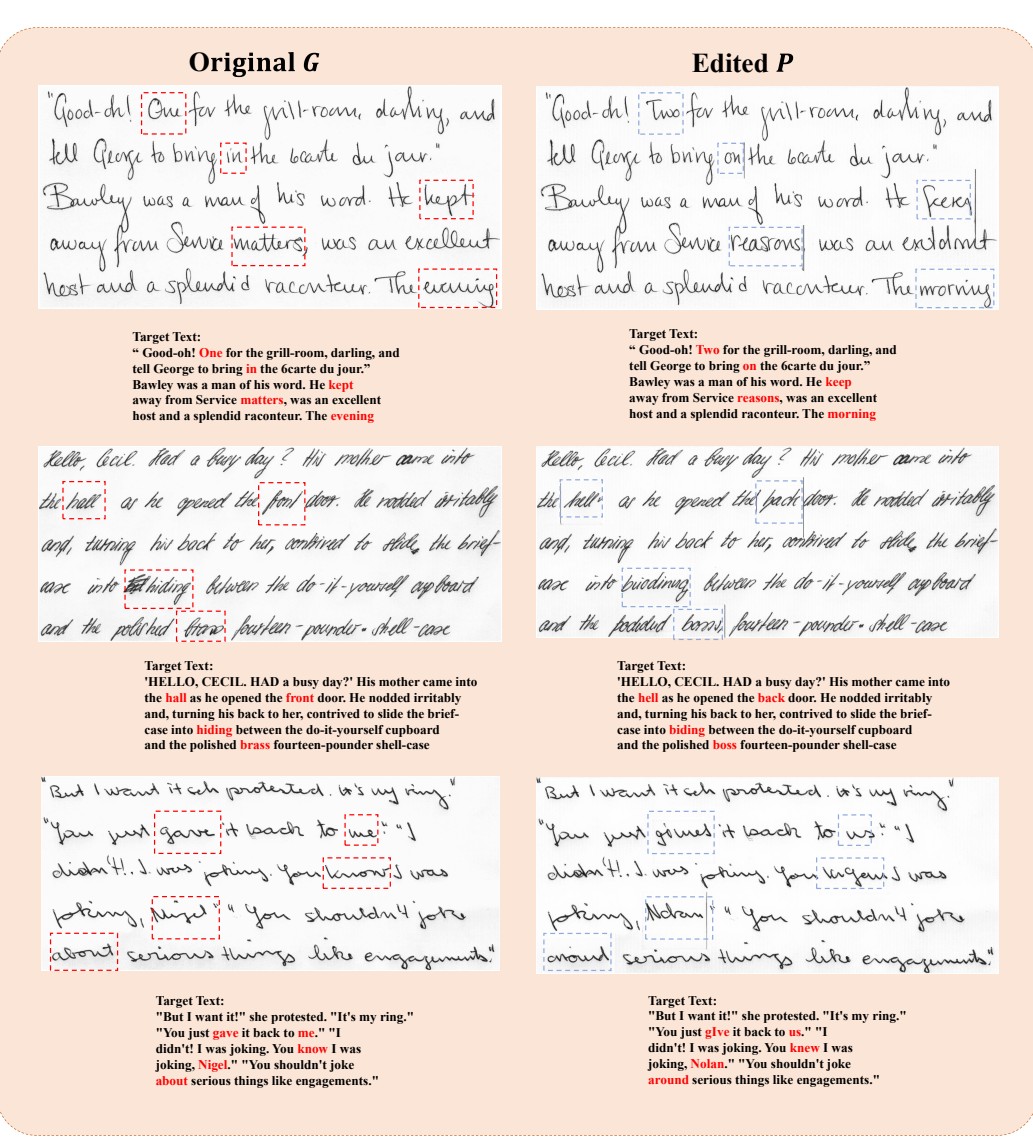

Figure 23: More examples of English paragraph editing.

Figure 24: More examples of Chinese paragraph editing.

