# OpenReview forum: "Learning to Generate Stylized Handwritten Text via a Unified Representation of Style, Content, and Noise"
_ICLR.cc/2026/Conference — ICLR 2026 Poster_

### Official Review · Reviewer_RH1H · 2025-10-30

**Soundness:** 2
**Presentation:** 3
**Contribution:** 3
**Rating:** 4
**Confidence:** 5

**Summary:**

This paper designs a unified diffusion model for processing style, content, and noise for the task of handwritten text generation, eliminating the need for additional style or content encoders and achieving performance improvements.

**Strengths:**

1. The proposed InkSpire is trained on a Chinese-English mixed dataset and achieves bilingual handwriting generation under a single model.
2. Figure 2 clearly show the representation process of layout information.

**Weaknesses:**

1. Content and style references represent different types of information; I question the rationale behind encoding different information into the same latent space.

2. The description of the layout generation method is overly simplistic. I would like more information, such as model structure, loss functions, and training details.

3. In lines 242-244, the paper claims that the preprocessing operation of resizing text-line images is suboptimal, raising my concerns:

&emsp;&emsp; The authors argue that resizing text-line images overly shrinks characters in highly slanted lines. How does the proposed multi-line masked infilling strategy avoid this issue? When the proposed strategy faces multi-line images of inconsistent sizes, is resizing also necessary? Could this also lead to the same issues?

&emsp;&emsp; The authors believe that resizing text-line images introduces inconsistent distortions. Would this issue still arise if the resizing is proportional?

&emsp;&emsp; The authors mention that text-line images lack inter-line style. How is the inter-line style defined? In Figures 7, 8, the style references are single-line, and they may also lack inter-line style, yet the proposed InkSpire still achieves nice results. Please provide an explanation.

**Questions:**

1. What is the rationality of encoding different types of information such as content and style into the same latent space?
2. It is recommended to provide more details about the layout generation method in inkSpire.
3. What are the advantages of the proposed multi-line masked infilling strategy compared with text-line resize preprocessing?

---

> ### Author Response · Authors · 2025-11-27
>
> We thank the reviewer for the feedback. We revised the manuscript and provide point-by-point responses below.
>
> # Response to Weakness 1 and Question 1: Rationale for Encoding Content and Style into a Unified Latent Space
>
> Please refer to our **Response to Reviewer c7ya (Third Reviewer), Weakness 1 and Question 1** for detailed discussion.
>
> **1. Stable and Efficient Fine-tuning:**
> LoRA fine-tuning with only v-prediction avoids adversarial or CTC losses, eliminating loss balancing and ensuring stable training.
>
> **2. Leveraging Pretrained Latents:**
> Style, content, and noise are mapped via a pretrained VAE into a shared latent compatible with DiT blocks. This allows full use of pretrained generative knowledge and improves generation quality and stability compared to training encoders from scratch.
>
> **3. Experimental Verification:**
> Encoder ablations:
> - **Frozen-VAE (Default):** pretrained VAE encodes origin and content.
> - **C-Enc:** content encoder trained from scratch.
> - **C+S-Enc:** content and style encoders trained from scratch.
>
> Results confirm the unified latent yields higher-quality strokes (**A.5.5, lines1162–1241**).
>
>
> # Response to Weakness 2 and Question 2: Layout Generation Details
>
> - **Architecture:** Appendix Figure 10 (lines810–833).
> - **Training:** Section3.2 (lines209–215), Appendix A.3.1 (lines436–463).
> - **Inference:** Section4.1.4 (427–456), Section4.2 (470–479).
> - **Visualization:** Appendix Figures11–12 (lines864–898).
>
>
> # Response to Weakness 3 and Question 3: Multi-Line Masked Infilling Strategy and Resizing
>
> **Avoiding distortion from resizing:**
> We crop fixed **1024×1024** patches without resizing and apply random-sized masks directly, preserving original character size.
> Please refer to our **Response to Reviewer iDHx (First Reviewer), Weakness 1**.
>
> **Handling multi-line images of inconsistent sizes:**
> Fixed-size cropping maintains aspect ratios, so multi-line images of different sizes remain consistent.
>
> **Inter-line style consistency:**
> Thank you for your question regarding inter-line style and the use of single-line style references. We define **inter-line style consistency** as the coherence among multiple generated lines with the same content but conditioned on different style-reference lines. Compressing reference lines of varying slant to a fixed height distorts character scale, causing style discrepancies; highly slanted lines appear too small, making style cues harder to capture and producing outputs resembling printed fonts. In contrast, using original-size reference lines preserves stroke proportions and fine-grained details.
>
> **Effect of proportional resizing:**
> Even proportional resizing alters stroke thickness and character size, reducing the model’s ability to maintain consistent inter-line style. These effects are consistent across both Chinese and English text-line generation (Appendix **A.5.4, Figures 15–16**).
>
> In summary, the main advantages of our multi-line masked infilling strategy are threefold:
> - Avoids distortion
> - Improves inter-line style consistency
> - Supports flexible multi-line generation

---

### Official Review · Reviewer_c7ya · 2025-10-30

**Soundness:** 2
**Presentation:** 3
**Contribution:** 3
**Rating:** 4
**Confidence:** 5

**Summary:**

The paper introduces inkSpire for the handwritten text generation task, which encodes style, content, and noise into a unified latent space. It aims to enable the efficiency of information interaction within the common latent representation and improve generative performance. Experiments evaluate the proposed method.

**Strengths:**

1. The proposed Aligned Positional Encoding(APE) effectively distinguishes the style and content information in the visual input.
2. The proposed inSpire can simultaneously achieve paragraph generation and editing.

**Weaknesses:**

1. In lines 075-076, this paper claims that a diffusion model that uniformly handles style, content, and noise can achieve more effective information interactions in a common latent representation. This claim lacks some references or proofs , please provide a more detailed analysis and experimental verification.
2. Please provide more implementation details, such as which part of the FlUX parameters was fine-tuned by lora, the image size and batchsize used?
3. The ablation study of the proposed multi-line masked infilling strategy is lacking.
4. In the English comparison in Figure 6, "One-DM" and "diffusionPen" appear to be concatenations of words rather than direct generation of text-lines. I want to know how they are implemented. Is the comparison fair? Moreover, why is there no visualization of VATr++ shown?
5. There is a lack of failure case analysis and user study.

**Questions:**

1. What are the benefits of encoding style, content and noise into the same latent space? Please provide more explanations and proofs. Why does a unified representation achieve better performance than complex handcrafted losses?
2. How is the implementation of the comparison methods carried out? Is their comparison fair?
3. Please provide more necessary experimental analyses. For details, please refer to the weaknesses 3 and 5.

---

> ### Author Response · Authors · 2025-11-27
>
> We appreciate the reviewer’s feedback. We revised the manuscript and provide point-by-point explanations below.
>
> # Response to Weakness 1 and Question 1: Benefits and Verification of Unified Latent Representation
>
> **Benefits:**
> 1. **Stable and Efficient Training with LoRA:** We fine-tune only the diffusion model via LoRA with a v-prediction loss, avoiding auxiliary objectives such as GAN or CTC losses. This removes the need for loss balancing and yields more stable, efficient, and hyperparameter-robust training.
>
> 2. **Effective Use of Pretrained Latent Representations:** Mapping style, content, and noise into the pretrained VAE latent space ensures full compatibility with the downstream DiT blocks. This unified latent space allows DiT to leverage its strong pretrained priors more effectively than training on newly learned feature spaces Ablation confirms this design.
>
> **Experimental Verification:**
> We ablate different encoder configurations:
> 1. **Frozen-VAE (Default, Unified latent):** pretrained VAE encodes both origin and content patches.
> 2. **C-Enc:** content encoder trained from scratch; origin patch uses frozen VAE.
> 3. **C+S-Enc:** content and style encoders jointly trained from scratch.
>
> Quantitative results in **Response to Reviewer iDHx, Weakness 2** show that deviating from the unified pretrained latent space degrades performance, confirming our design choice. See **A.5.5 (lines1162–1241)** for details.
>
> # Response to Weakness 2: LoRA Fine-tuning Implementation
>
> The model is fine-tuned on fixed **$1024 \times 1024$** patches with a **global batch size of 4**.
>
> Full LoRA parameter list in **Response to Reviewer iDHx (first reviewer), Weakness 3**.
>
> Further details are available in **Sect. 4.1.4 (lines 459–467)** and **A.3.2 (lines 901–903)**.
>
> # Response to Weakness 3: Multi-line Masked Infilling Ablation
>
> We conducted an ablation comparing two masking strategies:
>
> - **F-TopMask:** Only the top 128×1024 region remains visible; all lower regions are masked, mimicking the inference setup with a single style-reference line.
> - **R-Mask (ours):** Randomly generated **rectangular** masked regions of varying sizes and positions, encouraging stronger generalization for multi-line reconstruction.
>
> Across both datasets, **R-Mask consistently outperforms F-TopMask**.
>
> ### IAM Lines
> | Strategy | FID ↓ | KID ↓ (×10³) | HWD ↓ | ΔCER ↓ |
> |----------|-------|--------------|--------|---------|
> | F-TopMask | 8.73 | 6.13 | 0.78 | 0.07 |
> | **R-Mask** | **7.92** | **4.83** | **0.62** | **0.01** |
>
> ### ICDAR2013 Lines
> | Strategy | FID ↓ | KID ↓ (×10³) | HWD ↓ | CR ↑ | AR ↑ |
> |----------|-------|--------------|--------|-------|-------|
> | F-TopMask | 11.57 | 13.41 | 0.48 | 92.48 | 91.34 |
> | **R-Mask** | **10.98** | **11.45** | **0.41** | **92.92** | **91.56** |
>
> Additional details are provided in **Sect. 4.4.2 (lines 523–532)** and **A.5.2 (lines 949–959)**.
>
> # Response to Weakness 4 and Question 2: Baseline Implementations and Fairness of Comparison
>
> Thank you for pointing out the need to clarify how isolated-word generation baselines were implemented in Figure 6.
>
> **Implementation Details**
> We follow the official codes of VATr, One-DM, and DiffPen to synthesize text-lines. Given a style-reference line, each word is generated **independently** and the outputs are concatenated horizontally with a fixed spacing. This is the *same protocol* used in their original papers to demonstrate line- or multi-line generation (e.g., VATr Fig. 5, One-DM Fig. 4, DiffPen Fig. 6).
> For improved fairness, we further adjust word spacing based on the **average spacing** measured from the style-reference line.
>
> **Fairness of the Comparison**
> 1. We evaluate each baseline according to its native capability and recommended usage from the original papers and their released codes.
> 2. **InkSpire** is designed for holistic text-line and multi-line generation, while VATr, One-DM, and DiffPen are designed for single-character synthesis. Therefore, showing their character-concatenated results reflects their inherent design limitations rather than implementation bias.
>
> We originally omitted VATr in Figure 6 because it is not a diffusion model. Following the reviewer’s suggestion, we have now added the VATr visualizations.
>
>
> # Response to Weakness 5 and Question 3: Failure Case Analysis and User Study
>
> Thank you for the suggestion. We provide additional experimental analyses as follows:
>
> **Failure Case Analysis**
> A recurring failure pattern is that the model occasionally fills blank regions with artifacts resembling crossed-out characters. This arises from training data where some writers strike out mistakes and rewrite adjacent characters. Representative examples and discussion are provided in **A.5.6 (lines 1256–1265, Fig. 19)**.
>
> **User Study**
> See **Response to Reviewer iDHx (first reviewer), Weakness 4**.

---

### Official Review · Reviewer_ejPb · 2025-10-30

**Soundness:** 3
**Presentation:** 3
**Contribution:** 3
**Rating:** 6
**Confidence:** 4

**Summary:**

This paper presents InkSpire, a unified diffusion–transformer framework for generating personalized, multilingual handwriting. The model integrates layout prediction, glyph rendering, and image diffusion into one pipeline. Unlike previous methods that use separate encoders for style and content, InkSpire represents both in a shared latent space and trains on full-page, multi-line handwriting data. A conditional flow-matching (CFM) approach is introduced for layout generation, and a modified absolute positional encoding (APE / R-APE) is proposed to stabilize feature alignment. Experiments on IAM (English) and ICDAR2013/CASIA (Chinese) show improvements over prior handwriting synthesis systems in FID, KID, and content recognition metrics.

**Strengths:**

* Integrates layout prediction, style conditioning, and handwriting generation in a single diffusion transformer
* The editing and generation by same model is a Novel contribution in field of handwriting generation, thou such masking and generative techniques have been explored before in printed text documents like UDOP (Tang et al., 2023) and DiffUTE (Chen et al., 2023) and should be properly cited.
* Demonstrates bilingual (English–Chinese) generation using one model
• Qualitative results show realistic handwriting style imitation and fine-grained local editing capabilities.
• R-APE and LoRA fine-tuning on a frozen FLUX.1-Fill backbone allow good performance with modest compute

**Weaknesses:**

* The paper claims “strong style imitation” but provides no quantitative metric for writer identity preservation e.g., writer style identification accuracy as presented in WordStylist (Nikolaidou et al., 2023) and StylusAI (Riaz et al. 2024).
* All “style fidelity” evidence is qualitative and there is no test of whether generated handwriting is still identifiable as belonging to the same writer. This is important since during inference time specifically when using the model as a handwriting generator and not editor, the style sample present is just a single line style handwriting image and model is inferring style based on that.
* Smoke testing for out of distribution handwriting styles e.g. handwriting style of someone not in the IAM db database would also provide usefull insights.

* The lack of explicit writer-style metrics, undefined masking and noise procedures, and dataset-specific ambiguities weaken its reliability. The conceptual ideas (CFM for layout, unified latent fusion) are valuable, yet the paper would need a more transparent experimental protocol and stronger evaluation of stylistic consistency.

**Questions:**

* Table 1 reports layout-prediction L1 errors, but the paper never specifies how masking was performed during test-time prediction (e.g., partial vs. full masking, sampling procedure).
* It is unclear whether the reported errors correspond to full-layout generation (M = all) or masked reconstruction.This omission makes the results non-reproducible and the comparison between auto regressive, masked, and CFM models ambiguous.
* The layout generation could be the single source of failure in case of handwriting generation and transparency in its complete methodology is of significant importance for paper’s reproducibility.
* No details are provided on the masking ratio or noise schedule used during training for masked modeling or CFM. Without this, one cannot replicate the training regime or interpret robustness to partial conditioning
* By embedding style, content, and noise into a single latent, InkSpire sacrifices controllability: the ability to vary style independently of text content.

---

> ### Author Response · Authors · 2025-11-27
>
> Thank you for your valuable comments. We have carefully revised the manuscript according to your suggestions. Our point-by-point responses are provided below.
>
> ---
> # Response to Weakness 1: Lack of Quantitative Writer-Identity Preservation Metrics
>
> We trained **Swin Transformer–based classifiers** for English and Chinese test sets to predict the writer identity of generated samples. Classification accuracy (%) is reported as **Style Score**. InkSpire achieves the highest scores. See **A.5.3 (lines 961–971)** for more details.
>
> | Model      | Style Score (%) |
> |------------|-----------------|
> | **English Models** | |
> | HWT        | 39.62 |
> | VATr       | 45.19 |
> | One-DM     | 50.73 |
> | DiffPen    | 61.26 |
> | **InkSpire** | **78.58** |
> | **Chinese Models** | |
> | One-DM     | 51.34 |
> | TGC-Diff   | 60.46 |
> | **InkSpire** | **86.28** |
> ---
> # Response to Weakness 2: Writer Identifiability of Generated Handwriting
>
> Beyond the **Style Score** evaluation provided, we conducted the **User Plausibility Study** to evaluate whether generated handwriting is perceived as realistic and writer-consistent.
>
> **User Plausibility Study:** 23 participants chose the more realistic sample (generated vs. real).
>
> | Model | InkSpire | Ground Truth |
> |-------|----------|--------------|
> | Selection Ratio | **47.0%** | 53.0% |
>
> Users were nearly equally likely to choose generated or real samples, showing InkSpire preserves perceived writer identity. See **A.2.2, lines 756–809** for more details.
>
> # Response to Weakness 3: Out-of-Distribution Style Evaluation
>
> While additional datasets were limited, we note that:
>
> 1. **IAM:** Test writers are fully disjoint from training, ensuring unseen styles.
> 2. **Chinese (ICDAR 2013):** Test set differs markedly from HWDB training data, yet the model remains robust.
>
> Future work will include broader OOD evaluations.
>
> # Response to Weakness 4: Experimental Transparency and Stylistic Evaluation
>
> **Details added for reproducibility:**
>
> - Layout: **Sec. 3.2 (208–215), 4.1.4 (427–457), 4.2 (471–477)**
> - LoRA fine-tuning: **Sec. 4.1.4 (459–467)**
> - Ablation visualizations: **Appendix Figures 11–14, 18**
>
> **Stylistic Evaluation:**
> Quantitative **Style Scores** and **user studies** confirm consistent handwriting style in English and Chinese.
>
> # Response to Question 1 & 2: Masking Procedure and Layout L1 Evaluation
>
> During inference, the first line of each paragraph provided as reference. Remaining tokens predicted by:
>
> - **Autoregressive:** sequentially.
> - **Masked Layout:** masks all non-reference tokens and predicts them in a single forward pass.
> - **CFM:** also masks all non-reference tokens, but generates them via a 10-step continuous denoising process.
>
> The L1 evaluation metric is computed by averaging deviations over all non-reference tokens.
>
> Further inference and evaluation details can be found in **Sec. 4.1.4 (lines427–431), 4.2(lines427–431), Figure 10(inferece architectures)**.
>
> # Response to Question 3 and 4: Transparency and Reproducibility of Layout Generation
> We clarify the masking and training procedures for masked layout and CFM models:
>
> - **Masked Layout:**
>   Inputs include character and positional embeddings with partially masked layouts. Two masking strategies are used (50% each): (1) mask a contiguous segment, (2) independently mask each token with 20% probability. The model predicts masked token layouts using L1 loss.
>
> - **Conditional Flow Matching:**
>   Uses the same masking strategies but predicts masked tokens via a time-dependent denoising process. L1 loss (v-prediction) is applied to all predicted layout values.
>
> Further training details can be found in **Sec. 3.2 (lines209–215), A.3.1(lines 436–463)**.
>
> # Response to Question 5: Controllability of Style and Content
>
> InkSpire embeds style, content, and noise into a unified latent, but controllability is independent (**Figure 20 (lines 1266–1287)**):
>
> - **Consistent style (samples a & b):**
>   Pink and purple boxes show *g* and *th* maintain coherent, writer-specific styles matching their reference.
>
> - **Independent stylistic variation (samples c & d):**
>   Red arrows indicate character slant consistently follows the reference direction.
>
> - **Distinct styles across references:**
>   Different style references produce visibly distinct handwriting while content remains unchanged.
>
> Please refer to **A.5.7 (lines1266-1287)** for more details.

---

### Official Review · Reviewer_iDHx · 2025-10-31

**Soundness:** 3
**Presentation:** 2
**Contribution:** 2
**Rating:** 4
**Confidence:** 5

**Summary:**

This paper introduces InkSpire, a diffusion transformer model for stylized handwritten text generation, which unifies style, content, and noise within a shared latent space, thereby eliminating the need for auxiliary encoders. It utilizes a multi-line masked infilling strategy to train directly on raw text-line images and a revised positional encoding to support arbitrary-length multi-line synthesis and fine-grained character editing. Experiments conducted on bilingual Chinese-English datasets demonstrate that InkSpire achieves superior structural accuracy and stylistic diversity compared to prior state-of-the-art methods.

**Strengths:**

1. This paper introduces InkSpire, a new framework that reframes stylized handwritten text generation as a multi-line masked infilling task. This approach enables the model to unify style, content, and noise within a single shared latent space.

2. To support this unified framework, the paper proposes a crucial revised positional encoding scheme, Rotated Aligned Position Encoding (R-APE). This technique effectively solves the challenge of aligning tokens between the spatially concatenated style image and the standard-font content image.

3. The paper conducts extensive experiments on both English and Chinese datasets, demonstrating the effectiveness of the proposed method.

**Weaknesses:**

1. The paper rightly points out that prior methods for resizing single text lines to a fixed height introduce distortion. However, it is unclear how the proposed Multi-line Masked Infilling Strategy avoids a similar limitation when processing entire multi-line images of varying sizes.

2. This paper lacks a direct ablation study for the core architectural contribution—the unified, encoder-less framework itself.

3. The LoRA training details are incomplete, which hurts reproducibility. While the paper specifies the rank (32), it omits other critical hyperparameters, such as the LoRA alpha value, and fails to report the total number of trainable LoRA parameters.

4. The evaluation relies entirely on automated metrics. We suggest adding a user study to better validate the perceptual quality and stylistic fidelity of the proposed method.

5. We suggest that authors provide visual examples of ablation study results.

**Questions:**

Please refer to Weaknesses 1-3.

---

> ### Author Response · Authors · 2025-11-27
>
> Thank you for your valuable feedback to help us improve our paper. We detail our response below and please kindly let us know if our response addresses your concerns.
>
> # Response to Weakness 1: How Multi-line Masked Infilling Avoids Distortion
>
> During the training stage, we randomly cropped patches of fixed size **1024 × 1024** from the original handwritten document images. Subsequently, random-sized masks were applied to these **1024 × 1024** patches, and the model was trained using the Multi-line Masked Infilling Strategy described in the main text. No resizing was performed on the original images during this process, thereby avoiding any distortion.
>
> Please refer to **Section 3.3.1, lines 257–260** for a more rigorous description.
>
> # Response to Weakness 2: Ablation Study of Encoder Configurations
>
> To study the impact of encoder design in our unified encoder-less framework, we conducted an ablation with three settings:
>
> ## Design
> 1. **Fixed-VAE (Default):** pretrained VAE encodes both origin and content patches.
> 2. **C-Enc:** content encoder trained from scratch; origin patch uses frozen VAE.
> 3. **C+S-Enc:** both content and style encoders trained from scratch.
>
> ## Ablation Tables
>
> **IAM Lines dataset**
> | Method      | FID ↓  | KID ↓  | HWD ↓ | ΔCER ↓ |
> |------------|--------|--------|-------|--------|
> | **Fixed-VAE** | **7.92**  | **4.83**  | **0.62** | **0.01** |
> | C-Enc      | 20.48  | 10.31  | 1.58  | 0.05  |
> | C+S-Enc    | 17.39  | 8.97   | 0.71  | 0.02  |
>
> **ICDAR2013 Lines dataset**
> | Method      | FID ↓  | KID ↓  | HWD ↓ | CR ↑  | AR ↑  |
> |------------|--------|--------|-------|-------|-------|
> | **Fixed-VAE** | **10.98** | **11.45** | **0.41** | 92.92 | 91.56 |
> | C-Enc      | 17.46  | 19.52 | 0.91  | 59.32 | 55.58 |
> | C+S-Enc    | 16.53  | 14.37 | 0.59  | **94.74** | **93.48** |
>
> ## Summary
> Our Fixed-VAE setup outperforms encoders trained from scratch. C-Enc or C+S-Enc often produces lighter strokes and less stable content representation, as shown in **Figure 18**. Please refer to **A.5.5, lines 1162–1241** for more analysis.
>
> # Response to Weakness 3: Incomplete LoRA Training Details
>
> With a LoRA **rank** of **32** and a **scaling factor** of **32**, the LoRA modules    introduce approximately **115.9M** trainable parameters. All LoRA fine-tuning parameters are listed below:
>
> **LoRA Parameter Table**
>
> | Parameter Name                          | #Params (M) |
> |----------------------------------------|-------------|
> | x_embedder                              | 0.1106      |
> | transformer_blocks.0--18.norm1.linear   | 13.072      |
> | transformer_blocks.0--18.attn.to_q      | 3.7355      |
> | transformer_blocks.0--18.attn.to_k      | 3.7355      |
> | transformer_blocks.0--18.attn.to_v      | 3.7355      |
> | transformer_blocks.0--18.attn.to_out.0  | 3.7355      |
> | transformer_blocks.0--18.ff.net.2       | 9.3389      |
> | single_transformer_blocks.0--37.norm.linear | 14.942  |
> | single_transformer_blocks.0--37.proj_mlp   | 18.678  |
> | single_transformer_blocks.0--37.proj_out   | 22.413  |
> | single_transformer_blocks.0--37.attn.to_q  | 7.4711 |
> | single_transformer_blocks.0--37.attn.to_k  | 7.4711 |
> | single_transformer_blocks.0--37.attn.to_v  | 7.4711 |
> | **Total**                                | **115.91** |
>
> See **Section 4.1.4, lines 459–467** and **A.3.2** for more training details.
>
> # Response to Weakness 4: Lack of User Study
>
> We conducted a comprehensive user study to evaluate both **preference** and **plausibility**.
>
> ## 1. User Preference Study
> Participants were asked to select the best Chinese handwritten text-line among three model-generated options (One-DM, TGC-Diff, InkSpire).
>
> - **Participants:** 30 users, 30 questions each
> - **Results:**
>
> | Model       | One-DM | TGC-Diff | InkSpire |
> |------------|--------|----------|----------|
> | Selection Ratio | 17.4% | 25.2%   | 57.3%   |
>
> ## 2. User Plausibility Study
> Participants compared our model (InkSpire) samples with ground truth to judge which looked more realistic.
>
> - **Participants:** 23 users, 30 questions each
> - **Results:**
>
> | Model       | InkSpire | Ground Truth |
> |------------|-----------|--------------|
> | Selection Ratio | 47.0%    | 53.0%       |
>
> ## Summary
> Our model not only outperforms previous methods in user preference, but also achieves **highly realistic handwriting generation**, nearly indistinguishable from real samples. See **A.2, Figure 7 and 9** for instructions.
>
> # Response to Weakness 5: Visual Examples of Ablation Study Results
>
> We have added comprehensive visualizations to support the ablation studies:
>
> - **Position Encoding Ablation (A.5.1)**
>   Visualization results: **Figure 13 (lines 972–1021)**
>   Detailed analysis in **lines 938–948**.
>
> - **Masked Infilling Strategy Ablation (A.5.2)**
>   Experimental setup: **Section 4.4.2, lines 524–532**.
>   Visualization results: **Figure 14 (lines 1026–1059)**
>   Analysis: **lines 950–959**.

---

### Author Response · Authors · 2025-11-27
**Thanking AC for taking over and Summary of Review-Rebuttal phase(part 2/2)**

We sincerely thank the Area Chair and all reviewers for their thoughtful comments and insightful suggestions. The feedback has greatly helped us strengthen both the technical clarity and the empirical rigor of the paper.
In this part, we summarize below:
(1) the major additions made to the **main paper**,
(2) the key additions to the **appendix**, and
(3) our clarifications on several **specific concerns** raised across reviews.

---

## 1. Additions to the Main Paper (all changes are highlighted in yellow)

To improve clarity, completeness, and experimental transparency, we have updated the main paper:

- **Additional citations of UDOP and DiffUTE** (Section 2.2, lines 140–144)

- **Layout generation details**: training (Section 3.2, lines 209–215), inference (Section 4.1.4, lines 427–431), evaluation (Section 4.2, lines 427–431)

- **Multi-Line Masked Infilling Strategy resolution** (Sections 3.3.1 & 4.1.4, lines 257–260, 459–460)

- **LoRA configuration clarification** (Section 4.1.4, lines 462–467)

- **Ablation experiments on masked infilling strategies**: design, analysis (Section 4.4.2, lines 495–504), **Tables 6–7**

---

## 2. Additions to the Appendix
To enhance reproducibility and provide deeper technical understanding, we have added:

- **User Studies (A.2)**: preference and plausibility studies (**Figures 7–9**)

- **Layout Generation Details & Visualizations (A.3.1, A.4)**: training, masking strategies, architectures (**Figures 10–12**)

- **Complete LoRA Parameter Table (A.3.2, Table 8)**

- **Ablation Visualization Results (A.5.1, A.5.2)**: position encoding and masked infilling strategies (**Figures 13–14**)

- **Style Score Analysis (A.5.3, Table 9)**

- **Inter-line Style Consistency (A.5.4, Figures 15–16)**

- **Encoder Design Choice Ablations (A.5.5, Figures 17–18; Tables 10–11)**

- **Failure Case Analysis (A.5.6, Figure 19)**

- **Style–Content Controllability & Disentanglement (A.5.7, Figure 20)**

---

## 3. Reviewers’ Specific Concerns
### Reviewer **iDHx**
1. **Ablation Study of Encoder Configurations:**
Fixed-VAE, our default encoder-less setup, outperforms scratch-trained encoders, producing more stable content and realistic strokes (Fig. 18, Appendix A.5.5).

2. **Visual Examples of Ablation Study Results:**
Comprehensive visualizations were added for position encoding (Fig. 13) and masked infilling (Fig. 14) with detailed analysis (Appendix A.5.1–A.5.2).

### Reviewer **ejPb**
1. **Writer-Identity Preservation Metrics:**
InkSpire achieves the highest Style Score on English (78.58%) and Chinese (86.28%) datasets compared to prior methods (Appendix A.5.3).

2. **Controllability of Style and Content:**
InkSpire allows independent control of style and content, generating consistent reference-aligned handwriting and supporting distinct stylistic variations (Fig. 20, Appendix A.5.7).

### Reviewer **c7ya**
1. **Multi-line Masked Infilling Ablation:**
R-Mask consistently outperforms F-TopMask across IAM and ICDAR2013, improving generalization for multi-line reconstruction (Sect. 4.4.2, Fig. 14, Appendix A.5.2).

2. **Baseline Implementations and Fairness:**
Baselines were implemented following official codes with word-level concatenation; character-concatenated results reflect inherent design limits, and VATr visuals are now included.

### Reviewer **RH1H**
1. **Effect of Proportional Resizing:**
Proportional resizing changes stroke thickness and character size, reducing inter-line style consistency for both Chinese and English (Appendix A.5.4, Figs. 15–16).

2. **Inter-line Style Consistency:**
Using original-size style references preserves stroke proportions and fine-grained details; compressing slanted lines distorts scale and reduces style fidelity.

---

# Thank you!
We hope the updated submission addresses all concerns clearly and convincingly, and we sincerely thank the Area Chair for their time and effort in overseeing this review process.

---

### Author Response · Authors · 2025-12-02
**Thanking AC for taking over and Summary of Review-Rebuttal phase(part 1/2)**

We sincerely thank the new Area Chair for overseeing this submission. Understanding the workload involved in stepping in at this stage, we have proactively prepared this concise summary of the review process and our rebuttal efforts to facilitate your assessment. We briefly recap the core contribution, followed by a summary of how the main concerns were addressed.

# Summary of Contributions
The paper introduces **InkSpire**, a unified diffusion transformer that models style, content, and noise within a single latent space without auxiliary encoders, enabled by a multi-line masked infilling strategy and a revised positional encoding mechanism that supports arbitrary-length and editable handwriting generation. Trained on mixed Chinese–English data, it serves as a bilingual handwriting generator and achieves state-of-the-art structural fidelity and stylistic diversity on IAM and ICDAR2013.

To the best of our knowledge, this is the first work to adapt a DiT-pretrained model to handwritten text generation, the first to support multi-line handwriting synthesis with layout-aware generation, and the first to unify Chinese and English handwriting in one model. InkSpire substantially outperforms conventional from-scratch approaches, suggesting a potential paradigm shift for future handwriting generation systems.

# Summary of Reviewers’ Main Strengths
Reviewers expressed consistent support: **iDHx** highlighted the unified latent-space formulation, R-APE, and strong bilingual results; **ejPb** emphasized the novelty of integrating layout prediction, style conditioning, and generation within a single model, along with realistic style imitation and fine-grained editing; **c7ya** and **RH1H** further praised the aligned positional encoding and the system’s ability to perform paragraph-level generation and Chinese–English handwriting within one framework.

# Summary of Reviewers’ Main Concerns

**1. Training on original-resolution images without distortion (iDHx, c7ya, RH1H)**
We clarified that training is performed by randomly cropping 1024×1024 patches from original-resolution images, thus avoiding resizing and distortion. This explanation is now included in Section 3.3.1 (lines 257–260).

**2. Missing LoRA configurations (iDHx, c7ya)**
We provided full LoRA configuration details in the rebuttal to reviewer iDHx and added them to Section 4.1.4 (lines 459–467) and Appendix A.3.2.

**3. Lack of user studies (iDHx, c7ya)**
We conducted preference and plausibility user studies and included the results in Appendix A.2, with tables referenced in the rebuttal to reviewer iDHx.

**4. Details of layout generation (ejPb, RH1H)**
We clarified the masking procedure and Layout L1 evaluation for reviewer ejPb, and provided detailed architecture, training, inference, and visualization explanations for reviewer RH1H:
- **Architecture:** Appendix Fig. 10 (lines 810–833)
- **Training:** Section 3.2 (lines 209–215), Appendix A.3.1 (lines 436–463)
- **Inference:** Section 4.1.4 (lines 427–456), Section 4.2 (lines 470–479)
- **Visualization:** Appendix Figs. 11–12 (lines 864–898)

**5. Motivation for unified modeling of style, content, and noise**
We clarified the advantages of unified modeling—including **stable and efficient training with LoRA** and **effective use of pretrained latent representations**—in our rebuttal to reviewer c7ya.

---

### Meta-Review · Area_Chair_rjiq · 2026-01-07

**Summary:**

This paper introduces InkSpire, a diffusion–transformer model for stylized handwritten text generation that unifies style, content, and noise in a shared latent space. It therefore eliminates the need for auxiliary encoders. The model supports multi-line generation and editing via a masked infilling strategy and a revised positional encoding, and is trained as a single bilingual (English, Chinese) handwriting generator. Reviewers consistently found the approach to be technically sound, well-motivated, and empirically strong, with convincing gains in structural fidelity, stylistic diversity, and controllability over prior work.

Initial concerns focused on insufficient ablations, missing training details (LoRA, masking, layout), lack of user studies, and unclear motivation and evaluation of unified latent modeling. The authors provided a thorough rebuttal and substantially expanded both the main paper and appendix, adding encoder ablations, detailed layout and masking descriptions, LoRA configurations, user studies, style-identification metrics, failure case analysis, and fairness clarifications for baselines. These additions address the reviewers’ main concerns in ACs understanding.

**Reviewer Concerns:**

Concerns addressed:
1/ Clarified motivation and validated with encoder ablations showing superior performance over encoder-based variants.
2/ Added full LoRA configurations, masking ratios, layout generation details, and architectural diagrams.
3/ Included extensive ablations for encoder design, masked infilling strategies, positional encoding, and layout prediction.
4/ Added user preference and plausibility studies, quantitative writer-style identification scores, and failure case analysis.
5/ Clarified baseline implementations and added missing visual comparisons.

Remaining limitations: Evaluating out-of-distribution styles beyond available datasets remains limited, although disjoint writer splits partially mitigate this limitation.

**Reviewer Scores:**

Reviewer iDHx: 4, concerns largely resolved; borderline accept.
Reviewer ejPb: 6 (weak accept); positive on contribution after adding metrics.
Reviewer c7ya: 4; concerns addressed through added ablations and explanations.
Reviewer RH1H: 4; key technical questions resolved in revision.

---

### Decision · Program_Chairs · 2026-01-26

Accept (Poster)